# IN-CONTEXT REINFORCEMENT LEARNING WITH ALGORITHM DISTILLATION

**Michael Laskin**[*][†]**, Luyu Wang**[*]**, Junhyuk Oh, Emilio Parisotto, Stephen Spencer, Richie Steigerwald, DJ Strouse, Steven Hansen, Angelos Filos, Ethan Brooks, Maxime Gazeau, Himanshu Sahni, Satinder Singh, Volodymyr Mnih**[†]
DeepMind

## ABSTRACT

We propose Algorithm Distillation (AD), a method for distilling reinforcement learning (RL) algorithms into neural networks by modeling their training histories with a causal sequence model. Algorithm Distillation treats learning to reinforcement learn as an across-episode sequential prediction problem. A dataset of learning histories is generated by a source RL algorithm, and then a causal transformer is trained by autoregressively predicting actions given their preceding learning histories as context. Unlike sequential policy prediction architectures that distill post-learning or expert sequences, AD is able to improve its policy entirely in-context without updating its network parameters. We demonstrate that AD can reinforcement learn in-context in a variety of environments with sparse rewards, combinatorial task structure, and pixel-based observations, and find that AD learns a more data-efficient RL algorithm than the one that generated the source data.

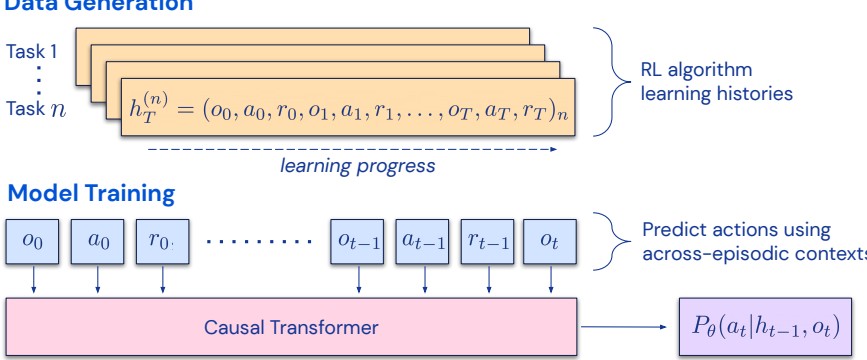

Figure 1: Algorithm Distillation (AD) has two steps – (i) a dataset of learning histories is collected from individual single-task RL algorithms solving different tasks; (ii) a causal transformer predicts actions from these histories using across-episodic contexts. Since the RL policy improves throughout the learning histories, by predicting actions accurately AD learns to output an improved policy relative to the one seen in its context. AD models state-action-reward tokens, and does not condition on returns.

## 1 INTRODUCTION

Transformers have emerged as powerful neural network architectures for sequence modeling (Vaswani et al., 2017). A striking property of pre-trained transformers is their ability to adapt to downstream tasks through prompt conditioning or in-context learning. After pre-training on large offline datasets, large transformers have been shown to generalize to downstream tasks in text completion (Brown et al., 2020), language understanding (Devlin et al., 2018), and image generation (Yu et al., 2022).

---
[*]Equal contribution.
[†]Corresponding authors: {mlaskin, vmnih}@deepmind.com.

Recent work demonstrated that transformers can also learn policies from offline data by treating offline Reinforcement Learning (RL) as a sequential prediction problem. While Chen et al. (2021) showed that transformers can learn single-task policies from offline RL data via imitation learning, subsequent work showed that transformers can also extract multi-task policies in both same-domain (Lee et al., 2022) and cross-domain settings (Reed et al., 2022). These works suggest a promising paradigm for extracting generalist multi-task policies – first collect a large and diverse dataset of environment interactions, then extract a policy from the data via sequential modeling. We refer to the family of approaches that learns policies from offline RL data via imitation learning as Offline Policy Distillation, or simply Policy Distillation[1] (PD).

Despite its simplicity and scalability, a substantial drawback of PD is that the resulting policy does not improve incrementally from additional interaction with the environment. For instance, the Multi-Game Decision Transformer (MGDT, Lee et al., 2022) learns a return-conditioned policy that plays many Atari games while Gato (Reed et al., 2022) learns a policy that solves tasks across diverse environments by inferring tasks through context, but neither method can improve its policy in-context through trial and error. MGDT adapts the transformer to new tasks by finetuning the model weights while Gato requires prompting with an expert demonstration to adapt to a new task. In short, Policy Distillation methods learn policies but not Reinforcement Learning algorithms.

We hypothesize that the reason Policy Distillation does not improve through trial and error is that it trains on data that does not show learning progress. Current methods either learn policies from data that contains no learning (*e.g.* by distilling fixed expert policies) or data with learning (*e.g.* the replay buffer of an RL agent) but with a context size that is too small to capture policy improvement.

Our key observation is that the sequential nature of learning within RL algorithm training could, in principle, make it possible to model the process of reinforcement learning itself as a causal sequence prediction problem. Specifically, if a transformer's context is long enough to include policy improvement due to learning updates it should be able to represent not only a fixed policy but a policy improvement operator by attending to states, actions and rewards from previous episodes. This opens the possibility that any RL algorithm can be distilled into a sufficiently powerful sequence model such as a transformer via imitation learning, converting it into an in-context RL algorithm. By in-context RL we mean that the transformer should improve its policy through trial and error within the environment by attending to its context, without updating its parameters.

We present Algorithm Distillation (AD), a method that learns an in-context policy improvement operator by optimizing a causal sequence prediction loss on the learning histories of an RL algorithm. AD has two components. First, a large multi-task dataset is generated by saving the training histories of an RL algorithm on many individual tasks. Next, a transformer models actions causally using the preceding learning history as its context. Since the policy improves throughout the course of training of the source RL algorithm, AD is forced to learn the improvement operator in order to accurately model the actions at any given point in the training history. Crucially, the transformer context size must be sufficiently large (*i.e.* across-episodic) to capture improvement in the training data. The full method is shown in Fig. 1.

We show that by imitating gradient-based RL algorithms using a causal transformer with sufficiently large contexts, AD can reinforcement learn new tasks entirely in-context. We evaluate AD across a number of partially observed environments that require exploration, including the pixel-based Watermaze (Morris, 1981) from DMLab (Beattie et al., 2016). We show that AD is capable of in-context exploration, temporal credit assignment, and generalization. We also show that AD learns a more data-efficient algorithm than the one that generated the source data for transformer training. To the best of our knowledge, AD is the first method to demonstrate in-context reinforcement learning via sequential modeling of offline data with an imitation loss.

## 2  BACKGROUND

**Partially Observable Markov Decision Processes:** A Markov Decision Process (MDP) consists of states $s \in \mathcal{S}$, actions $a \in \mathcal{A}$, rewards $r \in \mathcal{R}$, a discount factor $\gamma$, and a transition probability function $p(s_{t+1}|s_t, a_t)$, where $t$ is an integer denoting the timestep and $(\mathcal{S}, \mathcal{A})$ are state and action spaces. In

---

[1]What we refer to as Policy Distillation is similar to Rusu et al. (2016) but the policy is distilled from offline data, not a teacher network.

environments described by an MDP, at each timestep $t$ the agent observes the state $s_t$, selects an action $a_t \sim \pi(\cdot|s_t)$ from its policy, and then observes the next state $s_{t+1} \sim p(\cdot|s_t, a_t)$ sampled from the transition dynamics of the environment. In this work, we operate in the Partially Observable Markov Decision Process (POMDP) setting where instead of states $s \in S$ the agent receives observations $o \in \mathcal{O}$ that only have partial information about the true state of the environment. Full state information may be incomplete due to missing information about the goal in the environment, which the agent must instead infer through rewards with memory, or because the observations are pixel-based, or both.

**Online and Offline Reinforcement Learning:** Reinforcement Learning algorithms aim to maximize the return, defined as the cumulative sum of rewards $\sum_t \gamma^t r_t$, throughout an agent's lifetime or episode of training. RL algorithms broadly fall into two categories: on-policy algorithms (Williams, 1992) where the agent directly maximizes a Monte-Carlo estimate of the total returns or off-policy (Mnih et al., 2013; 2015) where an agent learns and maximizes a value function that approximates the total future return. Most RL algorithms maximize returns through trial-and-error by directly interacting with the environment. However, offline RL (Levine et al., 2020) has recently emerged as an alternate and often complementary paradigm for RL where an agent aims to extract return maximizing policies from offline data gathered by another agent. The offline dataset consists of $(s, a, r)$ tuples which are often used to train an off-policy agent, though other algorithms for extracting return maximizing policies from offline data are also possible.

**Self-Attention and Transformers** The self-attention (Vaswani et al., 2017) operation begins by projecting input data $X$ with three separate matrices onto $D$-dimensional vectors called queries $Q$, keys $K$, and values $V$. These vectors are then passed through the attention function:

$$\text{Attention}(Q, K, V) = \text{softmax}(QK^T/\sqrt{D})V. \qquad (1)$$

The $QK^T$ term computes an inner product between two projections of the input data $X$. The inner product is then normalized and projected back to a $D$-dimensional vector with the scaling term $V$. Transformers (Vaswani et al., 2017; Devlin et al., 2018; Brown et al., 2020) utilize self-attention as a core part of the architecture to process sequential data such as text sequences. Transformers are usually pre-trained with a self-supervised objective that predicts tokens within the sequential data. Common prediction tasks include predicting randomly masked out tokens (Devlin et al., 2018) or applying a causal mask and predicting the next token (Radford et al., 2018).

**Offline Policy Distillation:** We refer to the family of methods that treat offline Reinforcement Learning as a sequential prediction problem as Offline Policy Distillation, or Policy Distillation (PD) for brevity. Rather than learning a value function from offline data, PD extracts policies by predicting actions in the offline data (*i.e.* behavior cloning) with a sequence model and either return conditioning (Chen et al., 2021; Lee et al., 2022) or filtering out suboptimal data (Reed et al., 2022). Initially proposed to learn single-task policies (Chen et al., 2021; Janner et al., 2021), PD was recently extended to learn multi-task policies from diverse offline data (Lee et al., 2022; Reed et al., 2022).

**In-Context Learning:** In-context learning refers to the ability to infer tasks from context. For example, large language models like GPT-3 (Brown et al., 2020) or Gopher (Rae et al., 2021) can be directed at solving tasks such as text completion, code generation, and text summarization by specifying the task through language as a prompt. This ability to infer the task from prompt is often called in-context learning. We use the terms 'in-weights learning' and 'in-context learning' from prior work on sequence models (Brown et al., 2020; Chan et al., 2022) to distinguish between gradient-based learning with parameter updates and gradient-free learning from context, respectively.

## 3 METHOD

Over the course of its lifetime a capable reinforcement learning (RL) agent will exhibit complex behaviours, such as exploration, temporal credit assignment, and planning. Our key insight is that an agent's actions, regardless of the environment it inhabits, its internal structure, and implementation, can be viewed as a function of its past experience, which we refer to as its *history*. Formally, we write:

$$\mathcal{H} \ni h_t := (o_0, a_0, r_0, \ldots, o_{t-1}, a_{t-1}, r_{t-1}, o_t, a_t, r_t) = (o_{\leq t}, r_{\leq t}, a_{\leq t}) \qquad (2)$$

and we refer to a *long*[2] history-conditioned policy as an *algorithm*:

$$P : \mathcal{H} \cup \mathcal{O} \to \Delta(\mathcal{A}), \qquad (3)$$

---

[2]Long enough to span learning updates, *e.g.* across episodes.

where $\Delta(\mathcal{A})$ denotes the space of probability distributions over the action space $\mathcal{A}$. Eqn. (3) suggests that, similar to a policy, an algorithm can be unrolled in an environment to generate sequences of observations, rewards, and actions. For brevity, we denote the algorithm as $P$ and environment (*i.e.* task) as $\mathcal{M}$, such that the history of learning for any given task $\mathcal{M}$ is generated by the algorithm $P_\mathcal{M}$.

$$(O_0, A_0, R_0, \ldots, O_T, A_T, R_T) \sim P_\mathcal{M}. \tag{4}$$

Here, we're denoting random variables with uppercase Latin letters, *e.g* $O$, $A$, $R$, and their values with lowercase Latin letters, *e.g.* $o$, $a$, $r$. By viewing algorithms as long history-conditioned policies, we hypothesize that any algorithm that generated a set of learning histories can be distilled into a neural network by performing behavioral cloning over actions. Next, we present a method that, provided agents' lifetimes, learns a sequence model with behavioral cloning to map long histories to distributions over actions.

## 3.1 ALGORITHM DISTILLATION

Suppose the agents' lifetimes, which we also call *learning histories*, are generated by the source algorithm $P^{\text{source}}$ for many individual tasks $\{\mathcal{M}_n\}_{n=1}^N$, producing the dataset $\mathcal{D}$:

$$\mathcal{D} := \left\{ \left( o_0^{(n)}, a_0^{(n)}, r_0^{(n)}, \ldots, o_T^{(n)}, a_T^{(n)}, r_T^{(n)} \right) \sim P_{\mathcal{M}_n}^{\text{source}} \right\}_{n=1}^N. \tag{5}$$

Then we distill the source algorithm's behaviour into a sequence model that maps long histories to probabilities over actions with a negative log likelihood (NLL) loss and refer to this process as *algorithm distillation* (AD). In this work, we consider neural network models $P_\theta$ with parameters $\theta$ which we train by minimizing the following loss function:

$$\mathcal{L}(\theta) := - \sum_{n=1}^N \sum_{t=1}^{T-1} \log P_\theta(A = a_t^{(n)} | h_{t-1}^{(n)}, o_t^{(n)}). \tag{6}$$

Intuitively, a sequence model with *fixed* parameters that is trained with AD should amortise the source RL algorithm $P^{\text{source}}$ and by doing so exhibit similarly complex behaviours, such as exploration and temporal credit assignment. Since the RL policy improves throughout the learning history of the source algorithm, accurate action prediction requires the sequence model to not only infer the current policy from the preceding context but also infer the improved policy, therefore distilling the policy improvement operator.

---

**Algorithm 1** `Algorithm Distillation`

---

**Require:** Train $\{\mathcal{M}^{\text{train}}\}$ and test $\{\mathcal{M}^{\text{test}}\}$ tasks, observations $o \in \mathcal{O}$, actions $a \in \mathcal{A}$, and rewards $r \in \mathcal{R}$.
**Require:** Network parameters $\phi_i$ for $i = 1, \ldots, N$ source RL algorithms.
**Require:** Network parameters $\theta$ for a causal transformer $P_\theta$ that predicts actions.
**Require:** An empty buffer to store data $\mathcal{D}$.
 1: **for** $i = 1 \ldots N$ **do**                                                          ▷ Part 1: Dataset Generation
 2:     Sample a task $\mathcal{M}_i^{\text{train}}$ randomly from the train task distribution.
 3:     Train the source RL algorithm $\phi_i$ until it converges to the optimal policy.
 4:     Save the learning history $h_T^{(i)} = (o_0, a_0, r_0, \ldots, o_T, a_T, r_T)_i$ to the dataset $\mathcal{D} \leftarrow \mathcal{D} \cup h_T^{(i)}$.
 5: **end for**
 6: **while** $P_\theta$ not converged **do**                                                ▷ Part 2: Algorithm Distillation
 7:     Randomly sample a multi-episodic subsequence $\bar{h}_j^{(i)} = (o_j, a_j, r_j, \ldots, o_{j+c}, a_{j+c}, r_{j+c})_i$ of length $c$.
 8:     Autoregressively predict the actions with $P_\theta$ and compute the NLL loss in Eq. 6.
 9:     Backpropagate to update the transformer parameters.
10: **end while**
11: **for** $k = 1 \ldots M_{\text{seeds}}$ **do**                                           ▷ Part 3: Autoregressive Evaluation
12:     Sample a task $\mathcal{M}_k^{\text{test}}$ randomly from the test task distribution. Initialize empty context queue $C$.
13:     Unroll the transformer $P_\theta(\cdot | C)$ in the environment storing sequential transitions (*i.e.* histories) in $C$.
14:     Measure the return accumulated by the agent for each episode of evaluation.
15: **end for**

---

## 3.2 PRACTICAL IMPLEMENTATION

In practice, we implement AD as a two-step procedure. First, a dataset of learning histories is collected by running an individual gradient-based RL algorithm on many different tasks. Next, a

sequence model with multi-episodic context is trained to predict actions from the histories. We describe these two steps below and detail the full practical implementation in Algorithm 1.

**Dataset Generation:** A dataset of learning histories is collected by training $N$ individual single-task gradient-based RL algorithms. To prevent overfitting to any specific task during sequence model training, a task $\mathcal{M}$ is sampled randomly from a task distribution for each RL run. The data generation step is RL algorithm agnostic - any RL algorithm can be distilled. We show results distilling UCB exploration (Lai & Robbins, 1985), an on-policy actor-critic (Mnih et al., 2016), and an off-policy DQN (Mnih et al., 2013), in both distributed and single-stream settings. We denote the dataset of learning histories as $\mathcal{D}$ in Eq. 5.

**Training the Sequence Model:** Once a dataset of learning histories $\mathcal{D}$ is collected, a sequential prediction model is trained to predict actions given the preceding histories. We utilize the GPT (Radford et al., 2018) causal transformer model for sequential action prediction, but AD is compatible with any sequence model including RNNs (Williams & Zipser, 1989). For instance, we show in Appendix L that AD can also be achieved with an LSTM (Hochreiter & Schmidhuber, 1997), though less effectively than AD with causal transformers. Since causal transformer training and inference are quadratic in the sequence length, we sample across-episodic subsequences $\bar{h}_j = (o_j, r_j, a_j \ldots, o_{j+c}, r_{j+c}, a_{j+c})$ of length $c < T$ from $\mathcal{D}$ rather than training full histories.

## 4 EXPERIMENTAL SETUP

### 4.1 ENVIRONMENTS

To investigate the in-context RL capabilities of AD and the baselines (see next section), we focus on environments that cannot be solved through zero-shot generalization after pre-training. Specifically, we require that each environment supports many tasks, that the tasks cannot be inferred easily from the observation, and that episodes are short enough to feasibly train across-episodic causal transformers - for more details regarding environments see Appendix B. We list the evaluation environments below:

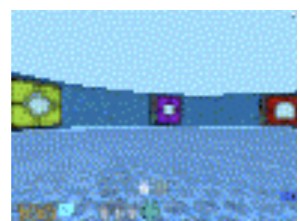

Figure 2: Agent view from the DM-Lab Watermaze environment. The task is to find a hidden platform that elevates once found.

**Adversarial Bandit:** a multi-armed bandit with 10 arms and 100 trials similar to the environment considered in RL$^2$ (Duan et al., 2016). However, during evaluation the reward is out of distribution. Reward is more likely distributed under odd arms 95% of the time during training. At evaluation, the opposite happens - reward appears more often under even arms 95% of the time.

**Dark Room:** a 2D discrete POMDP where an agent spawns in a room and must find a goal location. The agent only knows its own $(x, y)$ coordinates but does not know the goal location and must infer it from the reward. The room size is $9 \times 9$, the possible actions are one step left, right, up, down, and no-op, the episode length is 20, and the agent resets at the center of the map. We test two environment variants – Dark Room where the agent receives $r = 1$ every time the goal is reached and Dark Room Hard, a hard exploration variant with a $17 \times 17$ size and sparse reward ($r = 1$ exactly once for reaching the goal). When not $r = 1$, then $r = 0$.

**Dark Key-to-Door:** similar to Dark Room but this environment requires an agent to first find an invisible key upon which it receives a reward of $r = 1$ once and then open an invisible door upon which it receives a reward of $r = 1$ once again. Otherwise, the reward is $r = 0$. The room size is $9 \times 9$ making the task space combinatorial with $81^2 = 6561$ possible tasks. This environment is similar to the one considered in Chen et al. (2021) except the key and door are invisible and the reward is semi-sparse ($r = 1$ for both key and the door). The agent is randomly reset. The episode length is 50 steps.

**DMLab Watermaze:** a partially observable 3D visual DMLab environment based on the classic Morris Watermaze (Morris, 1981). The task is to navigate the water maze to find a randomly spawned trap-door. The maze walls have color patterns that can be used to remember the goal location. Observations are pixel images of size $72 \times 96 \times 3$. There are 8 possible actions in total, including going forward, backward, left, or right, rotating left or right, and rotating left or right while going forward. The episode length is 50, and the agent resets at the center of the map. Similar to Dark

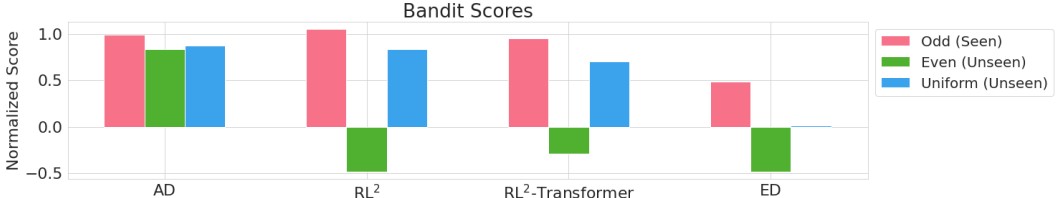

Figure 3: *Adversarial Bandit (Section 5):* AD, $RL^2$, and ED evaluated on a 10-arm bandit with 100 trials. The source data for AD comes from learning histories from UCB (Lai & Robbins, 1985). During training, the reward is distributed under odd arms 95% of the time and under even arms 95% of the time during evaluation. Both AD and $RL^2$ can in-context learn in-distribution tasks, but AD generalizes better out of distribution. Running $RL^2$ with a transformer generally doesn't offer an advantage over the original LSTM variant. ED performs poorly both in and out of distribution relative to AD and $RL^2$. Scores are normalized relative to UCB.

Room, the agent cannot see the location of the goal from the observations and must infer it through the reward of $r = 1$ if reached and $r = 0$ otherwise; however, the goal space is continuous and therefore there are an infinite number of goals.

## 4.2 BASELINES

The main aim of this work is to investigate to what extent AD reinforcement learns in-context relative to prior related work. AD is mostly closely related to Policy Distillation, where a policy is learned with a sequential model from offline interaction data. In-context online meta-RL is also related though not directly comparable to AD, since AD is an in-context *offline* meta-RL method. Still, we consider both types of baselines to better contextualize our work. For a more detailed discussion of these baseline choices we refer the reader to Appendix C. Our baselines include:

*Expert Distillation (ED):* this baseline is exactly the same as AD but the source data consists of expert trajectories only, rather than learning histories. ED is most similar to Gato (Reed et al., 2022) except ED models state-action-reward sequences like AD, while Gato models state-action sequences.

*Source Algorithm:* we compare AD to the gradient-based source RL algorithm that generates the training data for distillation. We include running the source algorithm from scratch as a baseline to compare the data-efficiency of in-context RL to the in-weights source algorithm.

*$RL^2$ (Duan et al., 2016):* an online meta-RL algorithm where exploration and fast in-context adaptation are learned jointly by maximizing a multi-episodic value function. $RL^2$ is not directly comparable to AD for similar reasons to why online and offline RL algorithms are not directly comparable – $RL^2$ gets to interact with the environment during training while AD does not. We use $RL^2$ asymptotic performance as an approximate upper bound for AD.

## 4.3 EVALUATION

After pre-training, the AD transformer $P_\theta$ can reinforcement learn in-context. Evaluation is exactly the same as with an in-weights RL algorithm except the learning happens entirely in-context without updating the transformer network parameters. Given an MDP (or POMDP), the transformer interacts with the environment and populates its own context (*i.e.* without demonstrations), where the context is a queue containing the last $c$ transitions. The transformer's performance is then evaluated in terms of its ability to maximize return. For all evaluation runs, we average results across 5 training seeds with 20 evaluation seeds each for a total of 100 seeds. A task $\mathcal{M}$ is sampled uniformly from the test task distribution and fixed for each evaluation seed. The aggregate statistics reported reflect multi-task performance. We evaluate for 1000 and 160 episodes for the Dark and Watermaze environments respectively and plot performance as a function of total environment steps at test-time.

## 5 EXPERIMENTS

The main research question of this work is whether an in-weights RL algorithm can be amortized into an in-context one via Algorithm Distillation. The in-context RL algorithm should behave in a similar

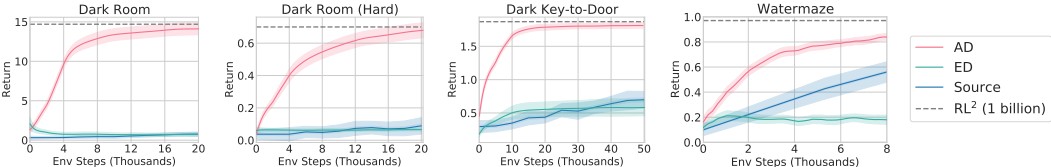

Figure 4: *Main results:* we evaluate AD, RL$^2$, ED, and the source algorithm on environments that require memory and exploration. In these environments, an agent must reach a goal location that can only be inferred through a binary reward. AD is consistently able to in-context reinforcement learn across all environments and is more data-efficient than the A3C ("Dark" environments) (Mnih et al., 2016) or DQN (Watermaze) (Mnih et al., 2013) source algorithm it distilled. We report the mean return $\pm$ 1 standard deviation over 5 training seeds with 20 test seeds each.

way as the in-weights one and exhibit exploration, credit assignment, and generalization capabilities. We begin our analysis in a clean and simple experimental setting where all three properties are required to solve the task - the Adversarial Bandit described in Sec. 4.

To generate the source data, we sample a set of training tasks $\{\mathcal{M}_j\}_{j=1}^N$, run the Upper Confidence Bound algorithm (Lai & Robbins, 1985), and save its learning histories. We then train a transformer to predict actions as described in Alg. 1. We evaluate AD, ED, and RL$^2$ and normalize their scores relative to UCB and a random policy $(r - r_{rand.})/(r_{UCB} - r_{rand.})$. The results are shown in Fig. 3. We find that both AD and RL$^2$ can reliably in-context learn tasks sampled from the training distribution while ED cannot, though ED does do better than random guessing when evaluated in-distribution. However, AD can also in-context learn to solve out of distribution tasks whereas the other methods cannot. This experiment shows that AD can explore the bandit arms, can assign credit by exploiting an arm once reached, and can generalize well out of distribution nearly as well as UCB. We now move beyond the bandit setting and investigate similar research questions in more challenging RL environments and present our results as answers to a series of research questions.

**Does Algorithm Distillation exhibit in-context reinforcement learning?** To answer this question, we first generate source data for Algorithm Distillation. In the Dark Room and Dark Key-to-Door environments we collect 2000 learning histories with an Asynchronous Advantage Actor-Critic (A3C) (Mnih et al., 2016) with 100 actors, while in DMLab Watermaze we collect 4000 learning histories with a distributed DQN with 16 parallel actors (see Appendix F for asymptotic learning curves of the source algorithm and Appendix O for hyperparameters). Shown in Fig. 4, AD in-context reinforcement learns in all of the environments. In contrast, ED fails to explore and learn in-context in most settings. We use RL$^2$ trained for 1 billion environment steps as a proxy for the upper bound of performance for a meta-RL method. RL$^2$ achieves a near-optimal asymptotic score in all the environments except for Dark Room (Hard). Despite learning from offline data, AD matches asymptotic RL$^2$ on the Dark environments and approaches it (within 13%) on Watermaze.

*Credit-assignment:* In Dark Room, the agent receives $r = 1$ each time it visits the goal location. Even though AD is trained to condition only on single timestep reward and not episodic return tokens, it is still able to maximize the reward, which suggests that AD has learned to do credit assignment.

*Exploration:* Dark Room (Hard) tests the agents exploration capability. Since the reward is sparse ($r = 1$ exactly once), most of the learning history has reward values of $r = 0$. Nevertheless, AD infers the goal from previous episodes in its context which means it has learned to explore and exploit.

*Generalization:* Dark Key-to-Door tests in-distribution generalization with a combinatorial task space. While the environment has a total of $\sim 6.5$k tasks, less than 2k were seen during training. During evaluation, AD both generalizes and achieves near-optimal performance on mostly unseen tasks.

**Can Algorithm Distillation learn from pixel-based observations?** DMLab Watermaze is a pixel-based environment that is larger than the Dark environments with tasks sampled from a continuous uniform distribution. The environment is partially observable in two ways - the goal is invisible until the agent has reached it and the first-person view limits the agent's field of vision. Shown in Fig. 4, AD maximizes the episodic return with in-context RL while ED does not learn.

**Can AD learn a more data-efficient RL algorithm than the one that produced the source data?** In Fig. 4, AD is significantly more data-efficient than the source algorithm. This gain is a byproduct of distilling a multi-stream algorithm into a single-stream one. The source algorithms

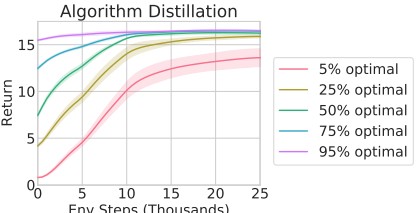 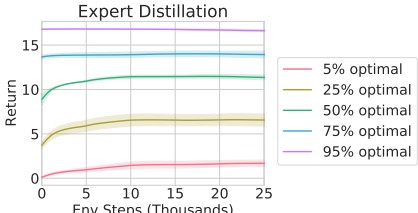

Figure 5: *AD and ED conditioned on partial demonstrations:* We compare the performance of AD and ED when prompted with a demonstration from the source algorithm's training history on Dark Room (semi-dense). While ED slightly improves and then maintains performance from the input policy, AD is able to improve it in-context until the policy is optimal or nearly optimal.

(A3C and DQN) are distributed, which means they run many actors in parallel to achieve good performance.[3] A distributed RL algorithm may not be very data-efficient in aggregate but each individual actor can be data-efficient. Since the learning history for each actor is saved separately, AD achieves similar performance to the multi-stream distributed RL algorithm, but is more data-efficient as a single-stream method.

These data-efficiency gains are also evident for distilling single-stream algorithms. In Fig 6 we show that by sub-sampling every $k$-th episode (where $k = 10$) from a single stream A3C learning history, AD can still learn a more data-efficient in-context RL algorithm (for more detail, see Appendix J). Therefore, AD can be more data-efficient than both a multi and single-stream source RL algorithm.

While AD is more data-efficient, the source algorithm achieves slightly higher asymptotic performance (see Appendix F). However, the source algorithm produces many single-task agents with a unique set of weights $\phi_n$ per task $\mathcal{M}_n$, while AD produces a single generalist agent with weights $\theta$ that are fixed across all tasks.

**Is it possible to accelerate AD by prompting it with demonstrations?** Although AD can reinforcement learn without relying on demonstrations, it has the added benefit that, unlike the source algorithm, it can be conditioned on or prompted with external data. To answer the research question, we sample policies from the hold-out test-set data along different points of the source algorithm history - from a near-random policy to a near-expert policy. We then pre-fill the context for both AD and ED with this policy data, and run both methods in the environment in Dark Room (Fig. 5). While ED maintains the performance of the input policy, AD improves every policy in-context until it is near-optimal. Importantly, the more optimal the input policy the faster AD improves it until it is optimal.

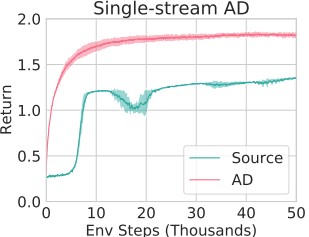

Figure 6: *Single-stream Algorithm Distillation:* AD trained on the learning history from an A3C agent with only one actor (*i.e.* single-stream). By training on subsampled learning histories (see Sec. 5), AD learns are more data-efficient in-context RL algorithm.

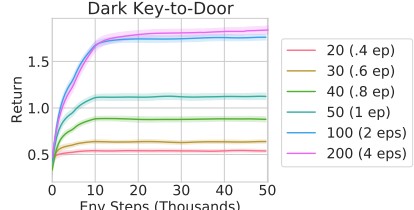

Figure 7: *Context size:* AD in Dark Key-to-Door with different context sizes. In-context RL only emerges once the context size is large enough and across-episodic.

**What context size is required for in-context RL to emerge?** We've hypothesized that AD requires sufficiently long (*i.e.* across-episodic) contexts to in-context reinforcement learn. We test this hypothesis by training several AD variants with different context sizes on the Dark Room environment. We plot the learning curves of these different variants in Fig. 7 and find that multi-episodic contexts of 2-4 episodes are necessary to learn a near-optimal in-context RL algorithm. Initial signs of in-context RL begin to emerge when the context size is roughly the length of an episode. The reason for this is likely that the context is large enough to retrain across-episodic information – *e.g.*, at the start of a new episode, the context will be filled with transitions from most of the previous episode.

[3]Indeed, current state-of-the-art RL algorithms such as MuZero (Schrittwieser et al., 2019) and Muesli (Hessel et al., 2021) rely on distributed actors.

## 6 RELATED WORK

**Offline Policy Distillation:** Most closely related to our work are the recent advances in learning policies from offline environment interaction data with transformers, which we have been referring to as Policy Distillation (PD). Initial PD architectures such as Decision Transformer (DT) (Chen et al., 2021) and Trajectory Transformer (Janner et al., 2021) showed that transformers can learn single-task policies from offline data. Subsequently the Multi-Game Decision Transformer (MGDT) (Lee et al., 2022) and Gato (Reed et al., 2022) showed that PD architectures can also learn multi-task same domain and cross-domain policies, respectively. Importantly, these prior methods use contexts substantially smaller than an episode length, which is likely the reason in-context RL was not observed in these works. Instead, they rely on alternate ways to adapt to new tasks - MGDT finetunes the model parameters while Gato gets prompted with expert demonstrations to adapt to downstream tasks. AD adapts in-context without finetuning and does not rely on demonstrations. A number of recent works have explored more generalized PD architectures (Furuta et al., 2021), prompt conditioning (Xu et al., 2022), and online gradient-based RL (Zheng et al., 2022). Some PD architectures such as DT and MGDT are instantiations of Upside Down RL (UDRL) Schmidhuber (2019); Srivastava et al. (2019) where rather than learning a value function, a policy is conditioned directly on the desired return. However, AD is not explicitly doing UDRL since it is not conditioned on returns. In fact, return maximization is not specified anywhere in the AD objective but rather emerges implicitly by modeling the learning histories of an RL algorithm.

**Meta Reinforcement Learning:** AD falls into the category of methods that learn to reinforcement learn, also known as meta-RL. Specifically, AD is an in-context offline meta-RL method. This general idea of learning the policy improvement process has a long history in reinforcement learning, but has been limited to meta-learning hyper-parameters[4] until recently (Ishii et al., 2002). In-context deep meta-RL methods introduced by Wang et al. (2016) and Duan et al. (2016) are usually trained in the online setting by maximizing multi-episodic value functions with memory-based architectures through environment interactions (Ni et al., 2022). Meta-RL through multi-episodic value functions has been done in both on-policy (Duan et al., 2016) and off-policy (Rakelly et al., 2019; Fakoor et al., 2020) settings. Another common approach to online meta-RL includes optimization-based methods that find good network parameter initializations for meta-RL (Hochreiter et al., 2001; Finn et al., 2017; Nichol et al., 2018) and adapt by taking additional gradient steps. Like other in-context meta-RL approaches, AD is gradient-free - it adapts to downstream tasks without updating its network parameters. Recent works have proposed learning to reinforcement learn from offline datasets, or offline meta-RL, using Bayesian RL (Dorfman et al., 2021) and optimization-based meta-RL (Mitchell et al., 2021). Given the difficulty of offline meta-RL, Pong et al. (2022) proposed a hybrid offline-online strategy for meta-RL.

**In-Context Learning with Transformers:** In this work, we make the distinction between in-context learning and incremental or in-context learning. In-context learning involves learning from a provided prompt or demonstration while incremental in-context learning involves learning from one's own behavior through trial and error. While many recent works have demonstrated the former, it is much less common to see methods that exhibit the latter. Arguably, the most impressive demonstrations of in-context learning to date have been shown in the text completion setting (Radford et al., 2018; Chen et al., 2020; Brown et al., 2020) through prompt conditioning. Similar methodology was recently extended to show powerful composability properties in text-conditioned image generation (Yu et al., 2022). Recent work showed that transformers can also learn simple algorithm classes, such as linear regression, in-context in a small-scale setting (Garg et al., 2022). Like prior in-context learning methods, Garg et al. (2022) required initializing the transformer prompt with expert examples. While the aforementioned approaches were examples of in-context learning, a recent work (Chen et al., 2022) demonstrated incremental in-context learning for hyperparameter optimization by treating hyperparameter optimization as a sequential prediction problem with a score function.

## 7 CONCLUSION

We have demonstrated that can distill RL algorithms by modeling their learning histories causally with imitation learning and that AD can learn more data-efficient algorithms than those that generated the source data. We hope that AD inspires further investigation into in-context reinforcement learning from the research community.

---

[4]For a recent example of meta-optimization in RL see Flennerhag et al. (2022).

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

## A   Discussion and Limitations

*Discussion*: While AD is more data-efficient than the source algorithm, we found that the source algorithm achieves a slightly higher asymptotic score in harder environments. This presents a trade-off between data-efficiency and asymptotic optimality when using AD that should be taken into consideration when applying AD to a specific problem. In terms of long-term consequences, AD presents a path for converting narrow single-task RL agents into multi-task generalist ones. To date, deep RL research has mostly focused on powerful single-task agents Mnih et al. (2015); Schrittwieser et al. (2019); Hessel et al. (2021). These algorithms have produced powerful but data-inefficient agents, which has limited their applicability beyond narrow domains. AD offers a path toward training substantially more data-efficient, though perhaps less optimal, generalist agents by distilling narrow RL algorithms into sequence models like transformers.

*Limitations*: While AD is a general method, in this work we've shown that AD generalizes to new tasks within the same domain. Showing the ability to distill cross-domain RL algorithms and generalize to new domains would be an interesting line if inquiry for future work. AD also requires storing many learning histories which could take up significant memory, though this may not be too much of an issue since the learning histories can be stored on disk rather than RAM. Perhaps the main limitation of AD is that most RL environments of interest have long episodes and modeling multi-episodic context requires more powerful long-horizon sequential models than the ones considered in this work. We believe this is a promising direction for future research.

## B   Environment Considerations

In this work, we consider environments where zero-shot generalization is difficult, so the agent must learn through trial and error. We also want environments where overfitting to any particular task is difficult to ensure our method is general. A final practical consideration is that we consider environments wher across-episodic histories can be feasibly modeled with a causal transformer. Given these considerations, our evaluation environments need to satisfy three criteria:

1. *Supports many tasks:* The environments must be multi-task to ensure that our agent and the baselines do not overfit to any single task and instead is able to in-context reinforcement learn across many tasks within a given domain.

2. *Task must be hard to infer:* To ensure that the downstream tasks are hard to generalize to in zero-shot, we use environments that require exploration. Namely, we require environments where either the task can only be inferred from the reward and not the observation, or tasks that are partially observable.

3. *Supports multi-episodic contexts:* Lastly, we impose a practical constraint - the environment episodes must be short enough such that a normal GPT-like transformer can fit multiple episodes in its context. Since this work introduces AD as a method, we wish to investigate it in the cleanest possible setting using a canonical architecture. We leave investigating AD with more complex architectures that scale to longer sequences for future work.

Prior related works (Chen et al., 2021; Lee et al., 2022; Janner et al., 2021; Reed et al., 2022) evaluated on Atari, OpenAI gym, and as well as other environments. However, Atari and OpenAI gym don't satisfy at least one of the above criteria. Atari and OpenAI gym episodes are often long and can contain thousands or more transitions per episode, so it's technically challenging to populate a causal transformer's context with across-episode histories. Indeed, the prior related works only considered within-episode context lengths. Additionally, it is often easy to infer the task from either the observation or the dense reward alone in both Atari and OpenAI gym, which reduces the need for exploration. For these reasons, we evaluate in environments that satisfy all three criteria instead.

## C   Closely Related Prior Methods

In our main results we use Expert Distillation (ED) as a baseline. Here, we discuss how the most closely related methods differ from AD and why ED is sufficient to support the paper's claims.

*Expert Distillation (ED):* ED is most similar to Gato (Reed et al., 2022), which models expert sequences from a converged RL policy using a causal transformer. ED also trains a causal transformer to predict actions using expert policy data. There are two key differences between ED and Gato. First, unlike Gato which utilizes small (relative to an episode length) within-episode contexts, ED is trained on the same across-episode contexts as AD, so the architectures used by ED and AD are the same. The benefits of AD cannot therefore be attributed to across-episode contexts alone but also learning progress in the offline data used to train AD. Second, ED models state-action-reward sequences while Gato models only state-action sequences. The main difference between ED and AD is that AD is trained on full multi-task learning histories rather than expert policy data.

*Decision Transformer (DT) (Chen et al., 2021) and Multi-Game Decision Transformer (Lee et al., 2022):* DTs learn return-conditioned policies from single-task offline data collected by an RL agent. While the training data itself (an RL agent's replay buffer) contains learning, the context sizes used in DT are too small to capture any learning progress or identify the task using across-episode information. For instance, the Atari experiments use a context of length $30 - 50$ tokens, or $10 - 17$ transitions. Atari games can have hundreds or thousands of transitions in a single episode, which means these contexts capture mostly within-episode information. Additionally, very little learning progress happens in the underlying replay buffer data within that many transitions.

Another difference between DT and AD / ED is that DT learns a return-conditioned model whereas AD / ED are both reward-conditioned. In our setting return-conditioning alone cannot yield an optimal policy since the agent does not know the task until after it explores the environment and can identify it using across-episode contexts. Since (i) DT uses small within-episodic contexts and (ii) return-conditioning would not help in the environments considered, this baseline is similar to ED with a small within-episode context which is strictly weaker than the long across-episode context variant of ED we consider.

*Trajectory Transformer (TTO) (Janner et al., 2021):* Like AD, TTO also models state-action-reward tokens but in addition to predicting actions it also learns a world model by predicting states and rewards. To maximize return, TTO then uses beam search to select high-reward actions. However, in our setting, TTO will run into the same problem as DT. To model rewards accurately it will need longer across-episodic contexts since one environment supports many tasks. Similar to DT, MGDT, and Gato, TTO uses smaller within-episode contexts. For this reason, TTO will fare no better than DT, MGDT, or ED in the settings we consider. We also note that in contrast to TTO, AD is model-free. In AD, actions are sampled from the transformer history-conditioned predictions and return maximization emerges from modeling the learning histories of an RL algorithm.

To summarize, AD differs from prior methods mainly because its context is across-episodic and hence large enough to capture learning progress and task information. AD could further be augmented by learning world models like TTO or conditioning on returns like DT, but these investigations would be well suited for future work since they are tangential to the main research question addressed in this work – whether in-context RL can emerge by imitating the learning histories of an RL algorithm with long across-episodic contexts.

AD is also closely related to prior work in in-context meta-RL. While both AD and in-context meta-RL model across-episodic histories with memory-based architectures, prior in-context meta-RL algorithms, such as RL2 (Duan et al., 2016) are trained online and rely on learning multi-episodic value functions with TD learning while AD is trained offline and uses a supervised imitation learning objective.

## D    EXPERT DISTILLATION MAIN RESULTS

We elaborate further on the main results in Fig. 4 and provide intuition regarding the behaviors of the ED baseline. In Dark Room, Dark Room (Hard), and Watermaze, ED performance is either flat or it degrades. The reason for this is that ED only saw expert trajectories during training, but during evaluation it needs to first explore (*i.e.* perform non-expert behavior) to identify the task. This required behavior is out-of-distribution for ED and for this reason it does not reinforcement learning in-context. In Dark Key-to-Door the agent is reset randomly at the beginning of each episode, whereas in all of the environments the agent's starting position is fixed. Due to random resets, the ED agent is sometimes reset by the first goal in Dark Key-to-Door which allows it to occasionally identify the first goal of the task, which is why it shows slight improvement.

## E    MODEL SIZE

We investigate how transformer capacity affects performance in Fig. 8. While in-context RL emerges across all model sizes investigated, we find that increasing the model depth, the model width in terms of embedding dimension, and (to a lesser extent) the number of attention heads improves performance on Dark Key-to-Door.

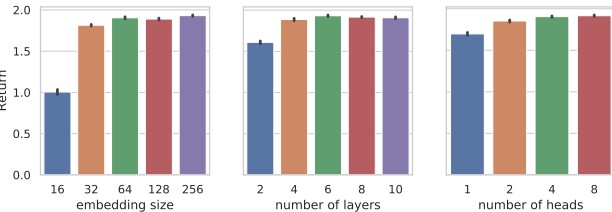

Figure 8: *Model size investigations:* We investigate how increasing model capacity affects AD. While in-context RL with AD emerges regardless of the model capacity, increasing the model depth and width helps improve AD until it achieves near-optimal performance.

# F    SOURCE ALGORITHM TRAINING RUNS

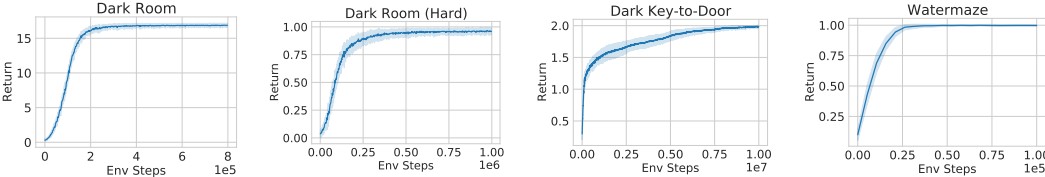

Figure 9: Asymptotic performance of the A3C (Mnih et al., 2016) and a Q-λ variant of the DQN (Mnih et al., 2013) RL algorithms used to produce learning histories for the Dark and Watermaze environments. These curves show the learning histories AD is trained on. The source algorithms plotted in Fig. 4 are the same as in these plots.

# G    LABEL SMOOTHING ABLATION

For the harder exploration task of Dark Room (Hard), we found that adding label smoothing regularization (Szegedy et al., 2016; Müller et al., 2019) improved the in-context learning ability of AD . In Figure 10 we ablate the benefit of using label smoothing for 3 different $\alpha$ values as well as with it turned off. Each curve in the figure denotes average performance over 5 training seeds. We can see that adding label smoothing up to a point improves the in-context learning ability of Algorithm Distillation, with performance continually increasing with the number of evaluation episodes.

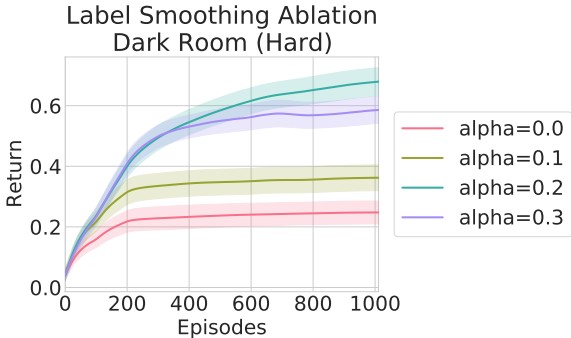

Figure 10: AD trained with different amounts of label smoothing on Dark Room (Hard).

# H    RL$^2$ NETWORK ARCHITECTURE: TRANSFORMER VS LSTM

We compared using a transformer as the architecture for RL$^2$ instead of an LSTM. In Figure 11, we ran both transformer and LSTM RL$^2$ agents over the Dark Room environment. The curves shown are the best from a sweep over learning rate and unroll length hyperparameters. The transformer architecture is 4-layers with a model size of 256 and pre-norm layer normalization placement. While both agents reached a similar level of final performance, all RL$^2$ transformer models trained tended to be more unstable with the average return not as consistent as with an LSTM architecture. Given the poor performance of the transformer-based RL$^2$ on the simpler Dark Room setting, our other experimental settings used the LSTM.

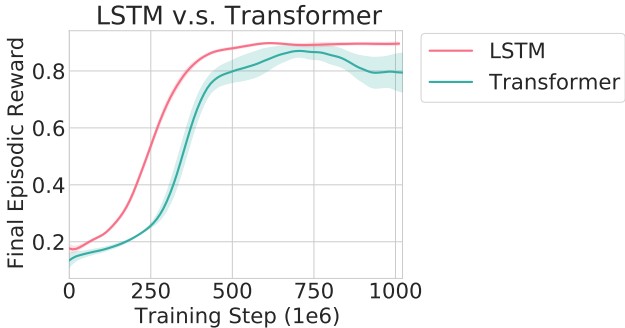

Figure 11: Comparison of LSTM and Transformer architecture for RL$^2$ agent on Dark Room. Each curve is averaged over 5 training seeds with the shaded area representing the standard error.

## I  NUMBER OF TRAINING TASKS IN SOURCE DATA

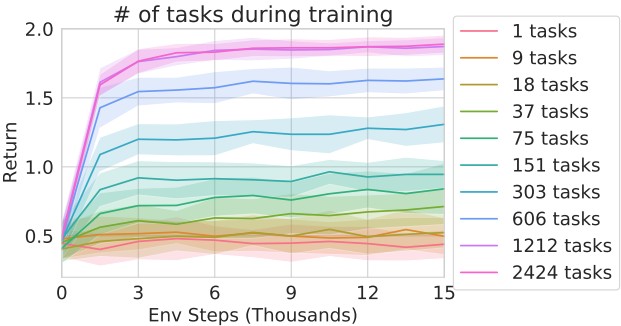

Figure 12: Algorithm distillation trained on different numbers of training tasks on Dark Key-to-Door evaluated on a fixed set of test tasks for 300 episodes of evaluation.

One interesting question is how many tasks AD needs to be trained on to learn an algorithm that generalizes to held out tasks. We trained AD on different numbers of Dark Key-to-Door training tasks and evaluated the resulting models on the same set of test tasks. Figure 12 shows the in-context learning plots for the resulting AD models on the set of test tasks. As a reminder, there are $81^2 = 6561$ unique Dark Key-to-Door tasks. Models trained on 1, 9 or 18 training tasks did not show any in-context learning on test tasks. While models trained on 37, 75 and 151 tasks did not achieve good performance overall, they did exhibit some in-context learning over the course of 300 episodes. The best models were trained on 1212 and 2424 tasks which corresponds to roughly $18\%$ and $37\%$ of the total number of tasks in the Dark Key-to-Door domain.

## J  SINGLE-STREAM ALGORITHM DISTILLATION

We provide more details around the experimental setup for the single-stream result shown in Fig. 6. We showed in Fig. 4 that when AD is trained on data from a subset of the actors of a distributed source RL algorithm, the resulting model is more data efficient than the source algorithm. Here we confirm that AD can produce a faster algorithm than the one it was trained on in the single-stream setting. For this experiment we trained A3C on 2048 Dark Key-to-Door tasks for 2000 episodes each. We then trained AD on the resulting data while subsampling the learning histories by a factor of 10. More concretely, we took every 10th episode from each of the learning history, which resulted in a 200 episode compressed learning trajectory for each task. Figure 6 compares the resulting AD model evaluated on a set of test tasks to the performance of the source algorithm on these tasks. The model learned by AD learns much faster than the source algorithm confirming that Algorithm Distillation can turn a slow gradient-based algorithm into a much more data efficient in-context learning algorithm.

# K    RANDOM MASK

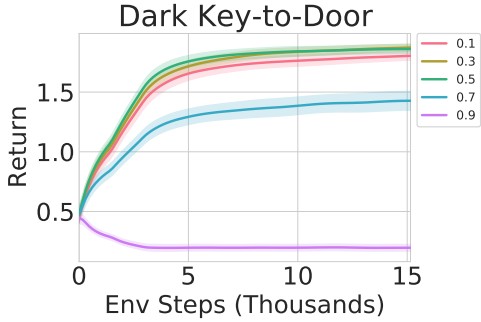

Figure 13: Downstream performance of Algorithm Distillation with different values of random masking during training in 9x9 1 goal gridworld.

During training, input tokens were randomly masked to avoid overfitting to training data. This plot shows the downstreams results on a 9x9 Dark Key-to-Door domain with different values of this random masking. Values of $0.3 - 0.5$ perform the best with the value of $0.3$ chosen for all experiments.

# L    AD NETWORK ARCHITECTURE: TRANSFORMER VS LSTM

Here we consider the importance of the Transformer architecture to the success of algorithm distillation (AD) by comparing to AD based off of an LSTM (Hochreiter & Schmidhuber, 1997). Specifically, the LSTM receives the concatenated embeddings of $(o_i, a_i, r_i)$ triplets up to the most recent time step $t - 1$. The output of the LSTM is then concatenated with the current observation $o_t$ embedding and both are then fed through a multi-layer perceptron (MLP) policy torso to produce a distribution over the present action $a_t$. The LSTM hidden size (512), MLP depth (2), and MLP width (256) were swept and tuned by grid search based on downstream reward attainment.

Comparing Transformer AD and LSTM AD on the Dark Key-to-Door task (Figure 14), we find that both agents are capable of in-context learning, demonstrating that the success of AD is not tied to the underlying network architecture. However, we also find that the Transformer variant consistently outperforms the LSTM variant, which is why all other experiments in this paper employ the Transformer variant. This finding is consistent with the recent wider success of Transformer-based architectures over recurrent neural network (RNN)-based architectures in sequence prediction tasks.

# M    DISTILLING ACTIONS VS PROBABILITIES

In the rest of the paper, we use (one-hot) actions taken by the source policy as the prediction target for AD. Here, we compare that choice to predicting the source policy probabilities from which that action was sampled. In other words, is it better to distill actions or probabilities?

Figure 15 compares all combinations of: 1) distilling actions vs probabilities, and 2) conditioning those predictions on past actions, probabilities, or both. The plot labels represent various input/output combinations, e.g. `sar->a` indicates observing states, actions, and rewards while predicting actions (i.e. the main variant of AD in the rest of the paper), `sar->p` represents using the same observations but instead predicting probabilities, and `sapr->a` represents observing states, actions, probabilities, and rewards while predicting actions, etc.

Two conclusions stand out from this plot. First, the original action distillation variant of AD (`sar->a`) performs best, followed closely by probability distillation (`sar->p`). In the paragraphs and experiments below, we explore why action distillation outperforms probability distillation. Second, all variants that condition on past probabilities catastrophically fail. We speculate on

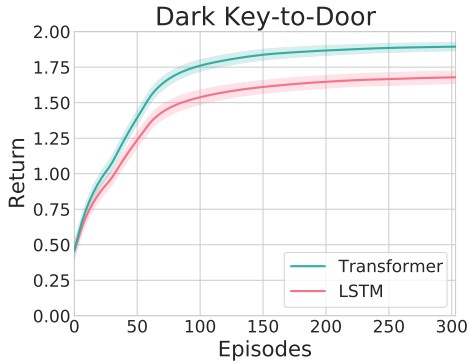

Figure 14: Comparison between algorithm distillation with a Transformer and LSTM architecture on Dark Key-to-Door. Mean ± 1 standard deviation over 5 training seeds and 20 evaluation seeds. 300 episodes corresponds to 15k environment steps.

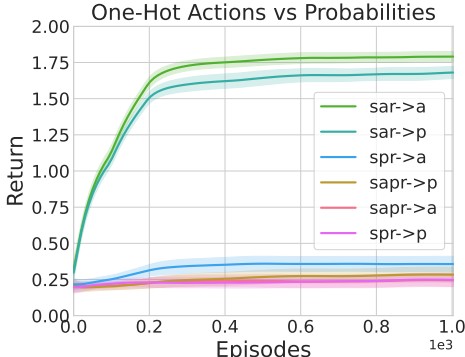

Figure 15: Comparison between algorithm distillation with various input/output combinations on Dark Key-to-Door. Mean ± 1 standard deviation over 5 training seeds and 20 evaluation seeds. 1000 episodes corresponds to 50k environment steps.

two somewhat contradictory reasons why this might occur: 1) The AD prediction task involves a combination of inferring the current policy, as well as predicting when policy updates will occur. Since observing policy probabilities provides more information about the current policy than actions do, it is possible that this leads to AD fully focusing on inferring the current policy and ignoring the prediction of policy updates. In other words, observing only actions taken may act as a useful information bottleneck. 2) On the other hand, observing probabilities may inadvertently leak information about when policy updates occur. If the transformer context includes multiple visits to the same state, AD could learn to compare the policy probabilities to infer whether a policy update occurred in between the two visits. If AD learned during training to rely on this information, then during autoregressive evaluation, it may be "waiting" for a policy update than never occurs. If this is the source of the issue, then a potential solution would be retraining with explicit policy update tokens and including them between evaluation episodes, however we leave this investigation to future work.

Now we return to the question of why action distillation (`sar->a`) outperforms probability distillation (`sar->p`). Intuitively, distilling one-hot actions will result in a more deterministic policy than distilling probabilities when training on finite data. Indeed, the red curves in the top two plots of Figure 16 show that probability distillation (top left) converges to a higher entropy (~1.2 bits) policy than does action distillation (top right, ~1 bit). Noting this discrepancy, we speculated that encouraging probability distillation towards a more deterministic policy might lead to increased performance. To do so, we experimented with an entropy penalty regularizer added to the AD loss

from equation 6, leading to the modified objective:

$$\tilde{\mathcal{L}}(\theta, \alpha) \coloneqq \sum_{n=1}^{N} \sum_{t=1}^{T-1} -\log P_\theta(A = a_t^{(n)} | h_{t-1}^{(n)}, o_t^{(n)}) + \alpha H\left[ P_\theta(A | h_{t-1}^{(n)}, o_t^{(n)}) \right], \tag{7}$$

where $H[\cdot]$ is the Shannon entropy in bits, and $\alpha$ is a regularization weight. The top two panels of Figure 16 show the effect of increasing $\alpha$ on decreasing policy entropy for action distillation (top right) and probability distillation (top left), throughout training. The bottom two panels show the corresponding changes in evaluation returns. Notably, entropy penalization indeed improves performance for probability distillation (bottom left), with the best performing regularization value ($\alpha = 0.1$) achieving a similar return to the original unregularized action distillation (bottom right, red). Interestingly, this is the amount of regularization that leads probability distillation to have the most similar entropy to the unregularized action distillation variant ($\sim$1 bit) as well. Further entropy penalization for action distillation, on the other hand, does not lead to increased performance (bottom right). Together, these results suggest that action distillation naturally leads to the "optimal" amount of entropy regularization on its own, at least in the environments we study.

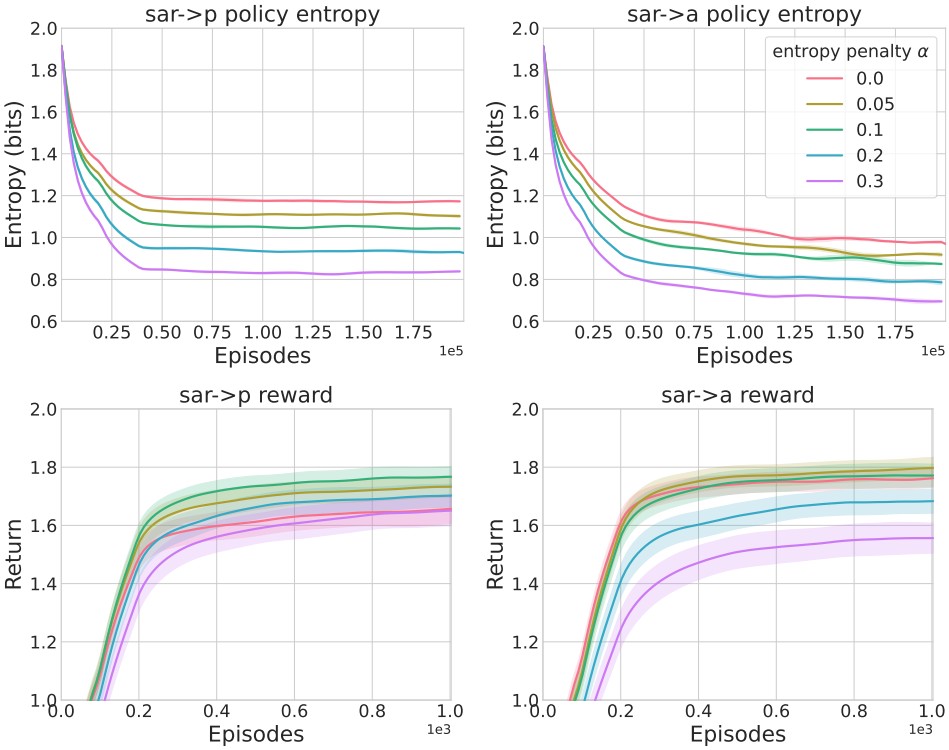

Figure 16: Policy entropy during training and reward during evaluation for the action distillation (`sar->a`) and probability distillation (`sar->p`) variants of AD on Dark Key-to-Door. Colors indicate the strength of the weight on the entropy penalty regularizer from equation 7. Mean $\pm$ 1 standard deviation over 5 training seeds and 20 evaluation seeds. 1000 episodes corresponds to 50k environment steps.

In addition to closing gap between probability and action distillation with an entropy penalty, we also explored retuning various hyperparameters for `sar->p` rather than reusing those tuned for `sar->a`. We found that many hyperparameters, such as the dropout rate, attention dropout rate, and sequence mask probability, had similar optimal values in the two cases, and so retuning them did not help. However, one hyperparameter that did have an effect was the size of the transformer context window (Figure 17). While action distillation performance increased only up to a context window size of 200 steps (4 episodes) and then plateaued (Figure 17, right), probability distillation performance continued to increase up to a context window size of 300 steps (6 episodes) before plateauing (Figure 17, left). Thus, while a comparison at our default context window size of 200

steps favored action distillation, for larger context window sizes, action and probability distillation performed similarly. We leave any further explanation of why probability distillation requires larger context windows than action distillation to future work.

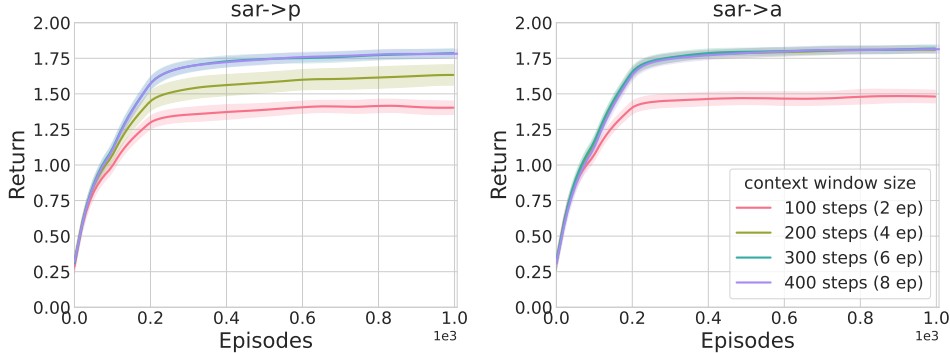

Figure 17: Context window size dependence for the action distillation (`sar->a`) and probability distillation (`sar->p`) variants of AD on Dark Key-to-Door. Mean $\pm$ 1 standard deviation over 5 training seeds and 20 evaluation seeds. 1000 episodes corresponds to 50k environment steps.

# N  ALGORITHM DISTILLATION HYPERPARAMETERS

| Hyperparameter | Dark Room | Dark Room (Hard) | Dark Key-to-Door | Watermaze |
|---|---|---|---|---|
| Embedding Dim. | | 64 | | |
| Number of Layers | | 4 | | |
| Number of Heads | | 4 | | |
| Feedforward Dim. | | 2048 | | |
| Position Encodings | | Absolute | | |
| Layer Norm Placement | | Post Norm | | |
| Dropout Rate | | 0.1 | | |
| Context Window | | 600 tokens (200 timesteps) | | |
| Attention Dropout Rate | 0.5 | 0 | 0.5 | 0.5 |
| Sequence Mask Prob | 0.3 | 0.5 | 0.3 | 0.3 |
| Label Smoothing $\alpha$ | 0 | 0.2 | 0 | 0 |

Table 1: Algorithm Distillation Architecture Hyperparameters.

| Hyperparameter | Value |
|---|---|
| Batch Size | 128 |
| Optimizer | Adam |
| $\beta_1$ | 0.9 |
| $\beta_2$ | 0.99 |
| Gradient Clip Norm Threshold | 1 |
| Learning Rate Schedule | Cosine Decay |
| Initial Value | 2e-6 |
| Peak Value | 3e-4 |

Table 2: Algorithm Distillation Optimization Hyperparameters.

| Layer | Hyperparameter | Value |
|---|---|---|
| *Conv Block* | | |
| Conv | Channel | 128 |
| | Kernel | 5 |
| | Stride | 2 |
| BatchNorm | Decay Rate | 0.999 |
| | eps | 1e-5 |
| Activation | - | ReLU |
| Max Pooling | Kernel | 2 |
| | Stride | 2 |
| Dropout | Rate | 0.2 |
| *Network* | | |
| Conv Blocks | - | 3 |
| Final Linear Layer | Units | 256 |

Table 3: Watermaze Image Encoder Hyperparameters.

# O   SOURCE RL ALGORITHM HYPERPARAMETERS

## O.1   DARK ENVIRONMENTS

| Hyperparameter | Value |
|---|---|
| Batch Size (Num. Actors) | 100 |
| $\lambda$ | 0.95 |
| Agent Discount | 0.99 |
| Entropy Bonus Weight | 0.01 |
| MLP Layers | 3 |
| MLP Hidden Dim | 128 |
| Optimizer | Adam |
| $\beta_1$ | 0.9 |
| $\beta_2$ | 0.999 |
| $\epsilon$ | 1e-6 |
| Learning Rate | 1e-4 |

Table 4: Source A3C Algorithm Hyperparameters for Dark Environments.

## O.2   DMLAB WATERMAZE

| Hyperparameter | Value |
|---|---|
| Batch Size | 8 |
| Rollout Length | 40 |
| Rollout Overlap | 31 |
| Number of Actors | 16 |
| Reply Buffer Capacity | 1e5 |
| Offline Data Fraction | 0.7 |
| $\lambda$ | 0.75 |
| $\epsilon$ | 0.01 |
| Agent Discount | 0.9 |
| Target Update Period | 50 |
| ResNet Channels | [32, 64, 64] |
| ResNet Kernels | [3, 3, 3] |
| ResNet Strides | [1, 1, 1] |
| Pool Kernels | [3, 3, 3] |
| Pool Strides | [2, 2, 2] |
| Optimizer | Adam |
| $\beta_1$ | 0.9 |
| $\beta_2$ | 0.999 |
| $\epsilon$ | 1e-6 |
| Gradient Clip Norm Threshold | 10 |
| Learning Rate | 1e-4 |

Table 5: Source DQN(Q-$\lambda$) Algorithm Hyperparameters for Watermaze.

## P RL$^2$ HYPERPARAMETERS

| Hyperparameter | Value |
| --- | --- |
| RL Algorithm | A3C |
| Learning Rate | 3e-4 |
| Batch Size | 256 |
| Unroll Length | 20 |
| LSTM Hidden Dim. | 256 |
| LSTM Number of Layers | 2 |
| Episodes Per Trial | 10 |

Table 6: RL$^2$ Hyperparameters used in "Dark" Environments.

| Hyperparameter | Value |
| --- | --- |
| RL Algorithm | DQN(Q-$\lambda$) |
| Learning Rate | 1e-4 |
| Batch Size | 96 |
| Unroll Length | 40 |
| LSTM Hidden Dim. | 256 |
| LSTM Number of Layers | 1 |
| Episodes Per Trial | 30 |

Table 7: RL$^2$ Hyperparameters used in the Watermaze Environment.

# Q ATTENTION MAPS

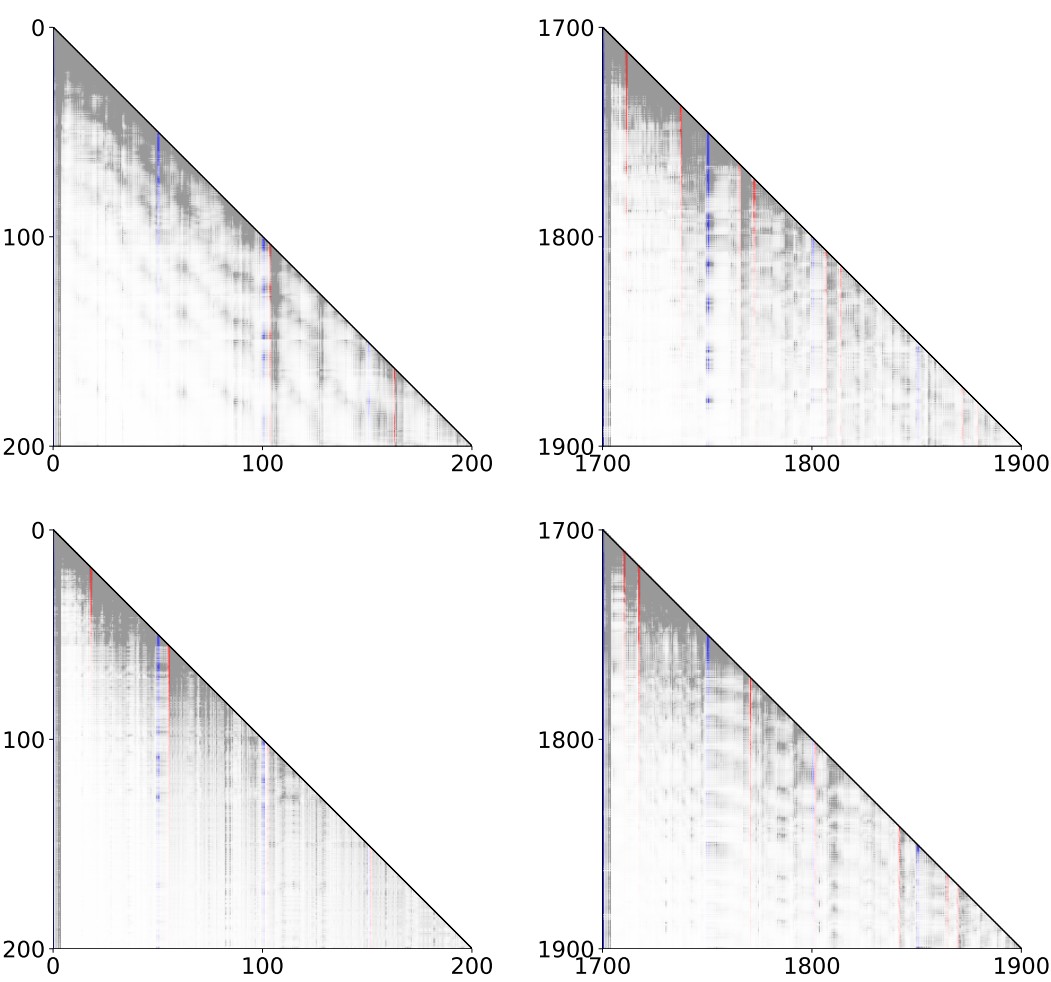

Figure 18: Attention maps for AD from five separate seeds. White and gray colors correspond to low and high attention. Red and blue colors indicate that those transitions correspond to an episode restart and a positive reward token, respectively. The left column plots attention for an AD transformer after 200 time-steps of evaluation (when the context is initially filled). The right column plots attention after 1900 steps (38 episodes) of evaluation. Each episode has a length of 50 steps. From these patterns, it is evident that AD attends to tokens across several episodes to predict its next action.

