# OpenReview forum: "In-context Reinforcement Learning with Algorithm Distillation"
_ICLR.cc/2023/Conference — ICLR 2023 notable top 5%_

### Official Review · Reviewer_8EkG · 2022-10-22

**Confidence:** 4
**Correctness:** 4
**Technical Novelty And Significance:** 3
**Empirical Novelty And Significance:** 4
**Recommendation:** 10

**Clarity, Quality, Novelty And Reproducibility:**

This paper is clear and, to the best of my knowledge, novel.  See above for more details on quality.

**Strength And Weaknesses:**

Edits:
- intro: “does not improve through trial an error”, “and” missing a “d”

Strengths:
- Well written; this paper was a pleasure to read.
- To the best of my knowledge, this work is novel.
- The fact that this approach worked was quite surprising to me.  This “surprise factor” makes this an important and interesting topic in my opinion.
- I also think that this kind of “in-context” learning (in other words, “learning” only within what-can-be-viewed-as-analogous-to the model’s short-term memory, without updating parameters) is an important thing to study. I hope this will inspire future work that better studies the intersection of this kind of learning with more traditional learning.

Weaknesses:
- The paper would greatly benefit by a more clear early definition of “in-context”, instead of making the reader infer it through context.  (There is sort of a definition in the abstract, but even that is more of an implied definition than a definition.  I suggest adding a more clear definition to the early introduction.)
- One might argue that the contribution is slightly limited: the authors simply propose a new idea and show that it works.  However, given the novelty and importance of this topic, I am not very concerned about the extent of the contribution.


**Summary Of The Paper:**

The authors propose AD.  AD works by training a transformer (or other causal sequence model) on RL histories of agents that are learning (exhibiting improving performance during the history).  Then after this training, when allowed to run on the environment, the transformer continues to improve performance, resulting in improving learning curves without any parameter updates.


**Summary Of The Review:**

This paper explores a topic that I believe is extremely important; the result (the fact that this approach worked) was surprising to me.  It is well-written, I have no significant concerns, and I have very little to say other than to recommend an accept.  I would consider recommending a “strong accept” if other reviewers felt similarly.

---

> ### Author Response · Authors · 2022-11-16
> **Thank you for your review**
>
> Thank you very much for reviewing our work. We appreciate that the reviewer found the method novel, results surprising, and the topic important.
>
> We fixed the typo the reviewer pointed out and respond to the suggestion made by the reviewer below:
>
> **Q1: The paper would greatly benefit by a more clear early definition of “in-context”, instead of making the reader infer it through context. (There is sort of a definition in the abstract, but even that is more of an implied definition than a definition. I suggest adding a more clear definition to the early introduction.)**
>
> Thank you for your feedback. Though we define in-context learning in the Background section, we agree that it would be clearer if we provided a short definition to the introduction. We have edited the text to incorporate the suggestion.

---

> > ### Comment · Reviewer_8EkG · 2022-11-18
> > **Update**
> >
> > After reading the other reviews (and responding to the authors' response to reviewer GFbm, see above), I have updated my confidence to a 4, and my recommendation to a 10.
> >
> > Reasoning: The other reviewers point out a few valid weaknesses, but I think this is a very important contribution overall, and that the most negative review is perhaps missing the core contribution.  Thus, I am updating my confidence and my score; I believe that this is a strong paper and an important contribution to the community.

---

### Official Review · Reviewer_E88f · 2022-10-23

**Confidence:** 5
**Correctness:** 4
**Technical Novelty And Significance:** 4
**Empirical Novelty And Significance:** 3
**Recommendation:** 8

**Clarity, Quality, Novelty And Reproducibility:**

The clarity of the paper is really good.

The quality of the research work is good, both in terms of experiment design and execution.

Despite its simplicity, the method presented in this paper is novel and interesting.

There are no clear reproducibility issues.

**Strength And Weaknesses:**

Strengths:
- The approach presented in this paper is at the same time very simple and intellectually stimulating;
- The empirical investigation is detailed and answers interesting questions one might have about this type of approaches;
- The presentation of the ideas is clear and concise.

Weaknesses:
- While I believe the careful investigation of such an idea is definitely worth even without an immediate implication on how to leverage it in a more standard context (i.e., when should I both to imitate my RL algorithm instead of simply running it?). It would be better to have a direct discussion on which ones would be the implication of improving this type of methods in the long run;
- Is the context length the only blocker in scaling this method to more complex tasks? From the current paper, it is not clear whether more complex transformer models (with longer context sizes) could actually be enough to scale to the usual reinforcement learning benchmarks.

**Summary Of The Paper:**

The paper proposes Algorithm Distillation, a transformer-based method for imitating the learning process implied by a reinforcement learning algorithm. The learning process is emulated by simply imitating the sequence of actions produced by an algorithm given a history of observations with the environment.

**Summary Of The Review:**

In short, I believe this is a well-executed investigation of a really intriguing idea. I recommend acceptance, but I encourage the authors to elaborate on the possible long-term consequences or more general implications of the development of such a class of methods.

---

> ### Author Response · Authors · 2022-11-16
> **Thank you for your review**
>
> Thank you for reviewing our paper and the positive feedback. We are glad that the reviewer found the method to be intellectually stimulating and the investigations to be detailed and interesting.
>
> We address the reviewer’s questions about the longer-term implications of our work below:
>
> **Q1: From the current paper, it is not clear whether more complex transformer models (with longer context sizes) could actually be enough to scale to the usual reinforcement learning benchmarks.**
>
> Indeed, most RL benchmarks such as Atari and Mujoco have long episodes (often 1k+ timesteps per episode). To model multi-episodic contexts we would need the sequence model to scale up to at least 6k tokens (1k timesteps per episode * 3 obs-action-reward per timestep * 2 episodes) and often even longer. Fortunately, there are recent architectures like TransformerXL and S4 that can feasibly model contexts of this length. For some RL environments, these models will be sufficient. However for others, and especially complex environments where episodes may span 100k-1m timesteps (e.g. Minecraft), new long-horizon architectures will likely need to be invented for a method like AD (or RL^2 for that matter) to scale effectively.
>
> **Q2: I encourage the authors to elaborate on the possible long-term consequences or more general implications of the development of such a class of methods.**
>
> Thank you for the suggestion - we have added a discussion section to the paper (Appendix A) that summarizes the following response. Perhaps the main long-term implication of Algorithm Distillation is a path from training narrow RL agents to generalist ones. To date, most RL breakthroughs have come from narrow agents, meaning that the algorithm produces one agent per task (DQN, AlphaZero, MuZero). AD shows that if we save the learning histories from narrow agents trained on many diverse tasks, then we can distill them into one generalist agent by converting the in-weight learning algorithm to an in-context one. Of course, prior methods exist that can also produce a single agent capable of in-context RL (e.g. RL^2), but AD offers a potentially scalable way of doing this from offline data.

---

### Official Review · Reviewer_HwSZ · 2022-10-28

**Confidence:** 4
**Correctness:** 3
**Technical Novelty And Significance:** 3
**Empirical Novelty And Significance:** 3
**Recommendation:** 6

**Clarity, Quality, Novelty And Reproducibility:**

- Quality and novelty
    - As argued in strengths&weaknesses, I think the novelty is not huge but is relevant enough. I do think the execution of this paper is very good, except for my concern on figure 4.
- Clarity
    - The paper in general reads very well. I think the connections to meta-RL comes a bit late, as I was initially confused on not seeing the reference to the field despite it being pretty clearly an offline meta-RL method. Also, as mentioned under weaknesses, a bit more care needs to be made to claim that AD distills DQN/PPO/A3C, as it probably only imitates them properly on nearby tasks.
    - One thing that wasn't clear to me was how much of the learning history needs to go into the context, and if we "skip" parts of it on purpose to try to learn faster or in order to fit in memory. I saw figure 7, but I wasn't sure whether that only meant at test time. If also at training time, how can it learn pretty well with only 1 episode of inner-loop experience?

**Strength And Weaknesses:**

- Strengths
    - I think the evaluation of this paper is very very good. The diversity of environments is great, and most of all, I was impressed by the many different questions the author asked themselves about the proposed approach, from out-of-distribution, to training on partial demonstrations or the importance of the length of the context. Great job!
    - The paper reads very clearly. In part this is because it sets out to do one job and do it well, and I appreciate that.
    - The idea is insightful and potentially useful in the long run for meta-RL.
- Weaknesses
    - I think the originality of the proposed approach is good enough to pass the bar for acceptance, but I wouldn't consider it great. It is relatively close to GATO and RL2. It is also close to multiple works that are not cited but very relevant: Bootstrapped Meta-learning [Flennerhag et al., Outstanding Paper award last ICLR] as well as Upside-Down RL [Schmidhuber] and, less closely, the learning-to-optimize community [Li&Malik '16, Andrychowicz et al. '16].
    - In multiple places, the comparison to the standard (non-learned) RL algorithm needs to be described more in favor of the latter rather than the proposed approach. For instance, it is argued that the "algorithm" coming from AD generalizes out-of-distribution, but this wouldn't apply to widely different tasks: an RL algorithm applies to image-input problems and torque-based inputs just as fine, but the algorithm from AD wouldn't have nearly that level of generality (and which other methods in meta-RL have started to achieve). Furthermore, the sentence "the source algorithm produces many single-task agents [...] while AD produces a single generalist agent" while technically true is quite incomplete, as AD needs to store each individual history to be able to represent each specialist agent.
    - A point that concerns me from the evaluation was that in the main figure, RL2 is shown only in its asymptotic value, as if it was an Oracle whose performance couldn't be put as just another curve (which may surpass AD). One may argue that it could be unfair for AD, as it is online rather than offline, but the same can be said about the Source being on a non-meta setting and not exploiting meta-training data, being at a disadvantage w.r.t. AD. I would really appreciate a clarification on this.

**Summary Of The Paper:**

This paper proposes to train a transformer to imitate offline data coming from a RL algorithm. When trained on a distribution of tasks, it can then generalise the behaviour of the RL algorithm into the new task. This is in comparison to recent papers using transformers to imitate already-trained RL policies, instead of algorithms. In this regard, it is closer to meta-RL algorithms like learning to reinforcement learn or RL^2, with the difference of being offline rather than online. The paper then evaluates on a wide variety of environments and analyses many different factors involved in the proposed approach.

**Summary Of The Review:**

The community will benefit from this paper as it is a very thorough exploration of a relevant idea, particularly in the current LLM-for-everything climate. The writing is also clear and it reads very well.

---

> ### Author Response · Authors · 2022-11-16
> **Thank you for your feedback and reviewing our work. We address the reviewer’s questions [Part 1 of 2]**
>
> Thank you for your thorough feedback and time spent reviewing our work. We really appreciate it. We are glad you found our evaluations interesting and extensive and the proposed method useful.
>
> We respond to the questions / clarifications you raised below:
>
> **Q1: It is relatively close to GATO and RL2. It is also close to multiple works that are not cited but very relevant: Bootstrapped Meta-learning [Flennerhag et al., Outstanding Paper award last ICLR] as well as Upside-Down RL [Schmidhuber]**
>
> Thank you for pointing these references out. We have added them to the Related Work section and discuss the differences between Algorithm Distillation (AD) and these approaches which we summarize below.
>
> Bootstrapped Meta-learning is a meta-optimization algorithm for RL which is different from the in-context meta-RL considered in RL^2 and our work. Bootstrapped meta-RL does meta-learning for hyperparameter optimization while AD and RL^2 meta-learn an RL algorithm in-context. These are two different types of meta-learning.
>
> While it appears similar, AD is not actually doing upside down RL. Upside down RL uses returns (or the desired behavior) as test-time inputs by conditioning the policy at test-time on its desired output. For example, Decision Transformers (DTs) do upside down RL by conditioning the policy to output maximal returns at test time. However, AD only conditions on obs-action-reward histories (and not returns!). So nowhere in the objective does AD know that it should maximize returns. Return maximizing behavior emerges implicitly by imitating the learning histories of an RL algorithm. That is, return maximization is baked into the data but not the training objective, so AD does not explicitly do upside down RL.
>
> We agree, however, these are interesting differences to note in the related work and we have added them to the text.
>
> **Q2: In multiple places, the comparison to the standard (non-learned) RL algorithm needs to be described more in favor of the latter rather than the proposed approach.**
>
> We appreciate your suggestion and have added a section (Appendix A) to incorporate the limitations of Algorithm Distillation as you suggested. We added text to the discussion section to address the specific points your raised in the following ways:
>
> *“AD generalizes out-of-distribution, but this wouldn't apply to widely different tasks.”* We’ve qualified that we’ve shown that AD generalizes out-of-distribution to tasks in the same domain, so that it is clear that we don’t mean to imply that AD generalizes to completely different tasks (e.g. DarkRoom to Atari generalization). In principle, AD is a general approach to offline meta-RL that could be coupled with a method like Gato to show cross-domain generalization but this would be an investigation for future work for scaling up AD since it would likely require a large number of diverse environments.
>
> *“AD needs to store each individual history to be able to represent each specialist agent.”* Indeed, AD needs sufficient task coverage (see Fig 12) to produce an in-context RL agent that can generalize well. However, the number of tasks required to generalize is smaller than the total number of possible tasks. For example, in Dark Key-to-Door which supports a total of 81^2=6561 tasks, we show that AD only needs to see ~1200 tasks (18% of total tasks) to learn a near-optimal in-context RL algorithm, and only ~100 tasks (<2% of total tasks) for signs of in-context RL to emerge. With that being said, we acknowledge the limitation you raised that AD needs to store many individual learning histories in the training data and have noted it in the limitations section (Appendix A).
>
> **Q3: Also, as mentioned under weaknesses, a bit more care needs to be made to claim that AD distills DQN/PPO/A3C, as it probably only imitates them properly on nearby tasks.**
>
> Thank you for the suggestion. Our claim is that AD distills the source RL algorithm in the domains considered, but we do not claim that it distills the algorithm optimally everywhere. For example, in Dark Room and Dark Key-to-Door AD learns a near-optimal RL algorithm so it is fair to say it distilled A3C there. However, in Dark Room (Hard) and Watermaze, the asymptotic performance of the source algorithm is slightly higher than that of AD so AD distilled the algorithm across most but not all tasks in those environments. However, AD has the benefit that it is substantially more data-efficient than the source algorithm, so it actually distilled a more data-efficient version of A3C / DQN (see Q4 for details). This presents a tradeoff between data-efficiency and asymptotic performance. While we noted this in the main results, we’ve added text to the discussion section to further highlight this tradeoff.
>
> [PART 1 of 2]

---

> > ### Author Response · Authors · 2022-11-16
> > **Thank you for your feedback and reviewing our work. We address the reviewer’s questions [Part 2 of 2]**
> >
> > **Q4: A point that concerns me from the evaluation was that in the main figure, RL2 is shown only in its asymptotic value, as if it was an Oracle whose performance couldn't be put as just another curve (which may surpass AD). One may argue that it could be unfair for AD, as it is online rather than offline, but the same can be said about the Source being on a non-meta setting and not exploiting meta-training data, being at a disadvantage w.r.t. AD. I would really appreciate a clarification on this.**
> >
> > **RL^2:** Indeed, the RL^2 baseline represents a proxy for upper bound performance in the same way that an online RL method (e.g. online DQN) could be used as an upper bound proxy for an offline RL method (e.g. CQL). We do not claim that AD outperforms RL^2 and in fact RL^2 performance is slightly higher than AD in the domains considered. The RL^2 asymptotic line is computed by running RL^2 in-context for the same number of evaluation timesteps as AD, after 1B steps of online pre-training. So RL^2 does surpass AD in these domains, but is online whereas AD is offline. This is not surprising since we would also expect an online DQN trained to convergence to outperform an offline DQN.
> >
> > **Source algorithm:** The reason we plot the source algorithm in Fig. 4 and Fig. 6 is that, in principle, if AD distilled the source algorithm perfectly then the AD curves and the source algorithm should be equally data-efficient. The meta-training data is just the learning histories of an RL algorithm and since AD is imitating these histories, it should not be any more data-efficient than the source algorithm itself, so the concern that access to the meta-training data benefits AD over the source algorithm is unlikely. If it imitates the meta-training data perfectly, the best AD should be able to do is as well as the source algorithm. Instead, we find that AD is much more data-efficient than the source algorithm, which we thought was an interesting observation to highlight to the reader.
> >
> > There are two concrete reasons that explain these data-efficiency gains; they occur when (a) AD distills a multi-stream algorithm and (b) when we subsample the training data. Please see the ablation “Can AD learn a more data-efficient RL algorithm than the one that produced the source data?” for further details. Note, however, that the source algorithm achieves better asymptotic performance after millions of environment steps in the harder environments (e.g. see DarkRoom Hard and Watermaze in Fig. 9).
> >
> > Finally, all of the environments considered in the paper have the property that it is impossible to know what the task is until you’ve encountered a reward (hence "Dark"), so it is impossible for AD (or RL^2 or the source algorithm) to exploit any task-specific information while it is exploring the environment and before it has encountered a reward.
> >
> > **Q5: One thing that wasn't clear to me was how much of the learning history needs to go into the context, and if we "skip" parts of it on purpose to try to learn faster or in order to fit in memory. I saw figure 7, but I wasn't sure whether that only meant at test time. If also at training time, how can it learn pretty well with only 1 episode of inner-loop experience?**
> >
> > Since Transformers have quadratic complexity in sequence length, we cannot fit the full learning history into the context and instead use a fixed size window that is sufficiently large to capture learning progress in the training data. Practically, this is done with random temporal cropping where we sample sequences of length $c$ from the training data. Also see “Training the Sequence Model” part of Section 3. The context window sizes have also been added to Appendix M Table 1.
> >
> > We use the same context window at train and test time, so Fig. 7 shows eval performance after training with a context windows of varying sizes. The reason that AD can still do pretty well with a context length of 1 episode is that it still contains cross-episodic information. Since we sample sequences randomly during training, a data sample with 1 episode worth of transitions often contains information from two neighboring episodes (e.g. last half of episode N and first half of episode N+1). Since the context is a queue it often contains cross-episodic information at both train and test time even if the context window is equal to or less than the length of an episode.
> >
> > Regarding “skipping” this is indeed one source of the data-efficiency gains. In figure 6, we show that AD trained with every k-th episode from the learning history (where k=10) is more data-efficient than the source algorithm. Another source of data-efficiency is distilling multi-stream algorithms which is what we do in Fig. 4 (see Q4 for more details).
> >
> > [PART 2 of 2]

---

### Official Review · Reviewer_GFbm · 2022-10-29

**Confidence:** 5
**Correctness:** 4
**Technical Novelty And Significance:** 2
**Empirical Novelty And Significance:** 2
**Recommendation:** 5

**Clarity, Quality, Novelty And Reproducibility:**

The paper reads well and it is clearly written. There are a good amount of experiments done in the paper except for the choice of baseline methods (see my comments in Strength And Weaknesses). Finally, as I mentioned in the previous section, the use of context for meta-RL is not something new and previous works studied that, though in online-RL.

**Strength And Weaknesses:**

While I find the result of this work interesting as it shows that it is possible to do more efficient offline meta-RL when the problem is formulated in the right way (e.g. using long context), I have a couple of concerns about this work:

- Meta-RL:
Using context/history in meta-RL has been studied in previous works [1,2,3] and all those works show that encoding past histories leads to significant improvement in performance (see results in [2]). Although the setting of those papers are online-RL, the similarity is uncanny and this work should have discussed those extensively in the paper (I'd suggest adding a section to discuss this). The only difference that I see with this paper and those works is use of transformer vs recurrent models to encode the context aside from online vs offline setting. Also, is it fair to say that the main contribution of this paper is to adopt the use of context/history in the decision transformer?

- Choice of baseline:
This paper uses RL^2 as a baseline. However, previous works [1,2] showed that RL^2 has poor performance in comparison to other meta-RL methods (see figure 2  of [1] ). Hence, choosing a better baseline method could provide a better conclusion about this paper.

- Improvement over behavioral policy:
Authors claim in multiple parts of the paper that "AD learns to output an improved policy relative to the one seen in its context." I am confused about this as I don't see the results support this claim. Take figure 4 as an example, AD doesn't outperform RL^2. Am I missing something here?

- Ablation studies:
"Can AD learn a more data-efficient RL algorithm than the one that produced the source data?" experiment shows that AD outperforms source algorithm. The setting of this experiment can be inconclusive as the source method and AD have different model sizes (?) and it appears to me AD uses larger network sizes. If that is the case, it is not a fair comparison. Moreover, "Training the Sequence Model" in page 5 can be a good experiment as long as it is a fair comparison in terms of model sizes. Finally, it would be useful if authors add a table in the appendix illustrating AD and baselines network/model sizes for each experiment in the paper to provide a better picture about the setting of the experiments.

- Context/History:
The paper says dataset D contains a set of "learning" histories. However lines 3 and 4 of Algorithm 1 show otherwise. In particular, Algorithm 1 shows that D contains ONLY expert data as optimal policies are used to collect data. This weakens the entire discussion of policy improvement in the paper as data contains only expert data.


[1] Efficient Off-Policy Meta-Reinforcement Learning via Probabilistic Context Variables, ICML 2019 http://proceedings.mlr.press/v97/rakelly19a/rakelly19a.pdf

[2] Meta-Q-Learning, ICLR 2020 https://openreview.net/pdf?id=SJeD3CEFPH

[3] Recurrent Model-Free RL Can Be a Strong Baseline for Many POMDPs, ICML 2022 https://proceedings.mlr.press/v162/ni22a/ni22a.pdf


**Summary Of The Paper:**

This paper proposes an offline meta-RL method that uses a causal transformer as a backbone model to predict actions . In particular, this method learns policies conditioned on a long context using a casual transformer where the context is built based on trajectories from multiple offline tasks. To show the effectiveness of their method, this paper runs various experiments on Adversarial Bandit, Dark Room, and DMLab Watermaze benchmarks.

**Summary Of The Review:**

This is an interesting paper and the results are promising. However, I'd make my final recommendation after rebuttal as I require some clarifications from others. In general, the proposed method is not original and based on previous meta-RL works. Using the decision transformer with context is the main contribution of this paper.

---

> ### Author Response · Authors · 2022-11-16
> **Thank you for reviewing our work. We address the reviewer’s questions and concerns [Part 1 of 3]**
>
> Thank you very much for reviewing our work. We appreciate the detailed questions and comments. It appears that there may have been a misunderstanding around the main idea of the paper, so we summarize our main contribution below and then elaborate by answering each question raised by the reviewer.
>
> Our main contribution is showing that in-context RL emerges via imitation learning of the learning histories of an RL algorithm with a sequence prediction model. We elaborate on the above by directly answering the reviewer’s questions:
>
> **Q1: The only difference that I see with this paper and [prior] works is use of transformer vs recurrent models to encode the context aside from online vs offline setting.**
>
> The above statement does not accurately describe the difference between Algorithm Distillation (AD) and prior work. The main difference between Algorithm Distillation (AD) and prior meta-RL approaches including the ones referenced by the reviewer [1,2,3] is that AD uses an imitation learning loss while prior approaches learn value functions using RL losses. Both use long contexts but AD is the first work to show that meta-RL can emerge from imitation learning if the training data consists of learning histories. We also show in Fig. 11 that AD works with both Transformers and LSTMs, so it is not transformer specific. AD is a new method - meta-RL via imitation learning of a learning algorithm - rather than a new architecture.
>
> Please also see other reviewers’ descriptions of AD which we believe accurately describe the method.
>
> *“This paper proposes to train a transformer to imitate offline data coming from a RL algorithm.”* Reviewer HwSZ
>
> *“The paper proposes Algorithm Distillation, a transformer-based method for imitating the learning process implied by a reinforcement learning algorithm.”* Reviewer E88f
>
> *“AD works by training a transformer (or other causal sequence model) on RL histories of agents that are learning (exhibiting improving performance during the history)”* Reviewer 8EkG
>
> **Q2: Although the setting of those papers are online-RL, the similarity is uncanny and this work should have discussed those extensively in the paper (I'd suggest adding a section to discuss this).**
>
> Thank you for the suggestion. We discuss related meta-RL approaches in detail in the Related Work (see the “Meta Reinforcement Learning” in the Related Work section). Thank you for pointing out these additional references which we have now added to the related work and edited the text to discuss their relevance to AD. As discussed in Q1, AD uses a different loss function (imitation) than prior methods and is trained offline whereas the references the reviewer provided use multi-episodic value functions and are trained online. Specifically, refs 1 & 2 mentioned by the reviewer show that meta-RL can be done online with off-policy RL, which was useful because prior approaches like RL^2 required an on-policy RL algorithm. The last ref [3] is a broad empirical investigation of how to correctly train RNN-based online RL algorithms that explicitly learn value functions, including online meta-RL with value functions as a subset of the investigation. AD is substantially different from these references - it uses a different loss function and is trained offline.
>
> **Q3: Also, is it fair to say that the main contribution of this paper is to adopt the use of context/history in the decision transformer?**
>
> We thank the reviewer for the clarification question. The above statement is not the main contribution. The main contribution is showing that in-context RL emerges if you model the learning histories of an RL algorithm with an imitation learning loss. In-context RL only emerges if the context is long enough, which we show in Fig. 7. So a long context is required for in-context RL to emerge but that is not the main idea of Algorithm Distillation. Please see Q1 for further discussion.
>
> Additionally, AD is not a Decision Transformer (DT). DTs explicitly estimate and condition on returns while AD does not. For details, please see Q8 and the Related Work for further details regarding the differences between AD and DT.
>
> [PART 1 of 3]

---

> > ### Author Response · Authors · 2022-11-16
> > **Thank you for reviewing our work. We address the reviewer’s questions and concerns [Part 2 of 3]**
> >
> > **Q4: Choice of baseline: This paper uses RL^2 as a baseline. However, previous works [1,2] showed that RL^2 has poor performance in comparison to other meta-RL methods (see figure 2  of [1] ). Hence, choosing a better baseline method could provide a better conclusion about this paper.**
> >
> > RL^2 achieves near-optimal performance on all environments except for DarkRoom (Hard), so a stronger baseline would not change the conclusion or claims of the paper. In our experiments we train RL^2 for 1B steps, by which point it has reached its asymptotic performance which is close to optimal in all but one of the environments. We added text that clarifies this in the Experiments section of the paper.
> >
> > **Q5: Improvement over behavioral policy: Authors claim in multiple parts of the paper that "AD learns to output an improved policy relative to the one seen in its context." I am confused about this as I don't see the results support this claim. Take figure 4 as an example, AD doesn't outperform RL^2. Am I missing something here?**
> >
> > The statement that "AD learns to output an improved policy relative to the one seen in its context" refers to AD doing in-context RL. It means that it can improve over the policy in its context (i.e. its own policy) during rollouts and is not a comparison to RL^2. In Fig. 4 AD is improving as a function of environment interactions and is doing so entirely in-context (no gradient updates or finetuning). AD starts with a bad policy with a near-zero score and outputs an improved policy with each episode (red curves in Fig. 4). Similarly, in Fig. 5 we show the behavior of AD when its context is pre-filled with interaction data from policies of varying optimality - from 5% optimal to 95% optimal. In Fig. 5, AD starts at the performance level of the input policy and then improves it in-context until it is optimal or near-optimal. Expert Distillation (ED) on the other hand can only maintain the performance of the input policy and cannot improve it. Finally, RL^2 is a baseline. AD does not need to outperform RL^2 for the statement that AD does policy improvement in-context to be true.
> >
> > **Q6: Ablation studies: "Can AD learn a more data-efficient RL algorithm than the one that produced the source data?" experiment shows that AD outperforms source algorithm. The setting of this experiment can be inconclusive as the source method and AD have different model sizes (?) and it appears to me AD uses larger network sizes. If that is the case, it is not a fair comparison. Moreover, "Training the Sequence Model" in page 5 can be a good experiment as long as it is a fair comparison in terms of model sizes. Finally, it would be useful if authors add a table in the appendix illustrating AD and baselines network/model sizes for each experiment in the paper to provide a better picture about the setting of the experiments.**
> >
> > Higher network capacity is not the reason AD is more data-efficient than the source algorithm. Recall that AD is doing imitation learning of the source algorithm’s learning data. In principle, with infinite capacity and training data AD should perfectly recover the source algorithm and match its data-efficiency regardless of the source algorithm’s network size. Additionally, since AD is doing imitation learning of the source algorithm, any gains due to using larger networks in the source algorithm would also transfer to AD. For these reasons higher capacity would not explain the difference in data-efficiency. We provide concrete reasons why AD is more data-efficient in the ablation study the reviewer referenced. Data-efficiency gains occur when (a) AD distills a multi-stream algorithm and (b) when we subsample the training data. Please see the ablation “Can AD learn a more data-efficient RL algorithm than the one that produced the source data?” for further details.
> >
> > Please see the tables in appendices M, N, and O for network hyperparameters for each experiment, which determine the network sizes.
> >
> > As a side note, the source algorithm actually uses more parameters in aggregate than AD. The source algorithm learns a unique set of parameters per task, while AD learns one set of parameters for all tasks. This means that the source algorithm uses in total `N = source_model_size * num_training_tasks` total parameters while AD uses `N = ad_model_size parameters`. As a result, the memory footprint of the source algorithm is much higher than that of AD since it learns a unique set of parameters for each task. For example, for Dark Key-to-Door the number of parameters needed to generate the training data from the source algorithm is ~100M, several times more than the AD transformer.
> >
> > [PART 2 of 3]

---

> > > ### Author Response · Authors · 2022-11-16
> > > **Thank you for reviewing our work. We address the reviewer’s questions and concerns [Part 3 of 3]**
> > >
> > > **Q7: Context/History: The paper says dataset D contains a set of "learning" histories. However lines 3 and 4 of Algorithm 1 show otherwise. In particular, Algorithm 1 shows that D contains ONLY expert data as optimal policies are used to collect data. This weakens the entire discussion of policy improvement in the paper as data contains only expert data.**
> > >
> > > We are thankful for the reviewer’s thorough engagement with our work. However, the above statement and specifically “Algorithm 1 shows that D contains ONLY expert data as optimal policies” are incorrect. In Algorithm 1, line 3 says that you run a source RL algorithm until it converges. Line 4 says that you “save the learning history” which includes all of the timesteps, from timestep 0 when the policy is random to timestep T when the algorithm has converged, not only the expert ones. That is, line 3 specifies when to stop the source RL training run, and line 4 says to keep the entire history of learning. AD is trained on data from the entire learning history. Expert Distillation (ED) is trained only on expert policy data.
> > >
> > > **Q8: Using the decision transformer with context is the main contribution of this paper.**
> > >
> > > Our main contribution is not that AD is DT with longer contexts but rather the observation that in-context RL emerges from modeling the learning histories of an RL algorithm with an imitation learning loss. For elaboration on this point please see Q1 and Q3.
> > >
> > > AD is not a Decision Transformer (DT), even though it appears similar. The key difference is that DTs explicitly condition on returns (the sum of rewards) during training, and then condition on the maximal return during eval. AD, on the other hand, does not condition on returns at all. AD conditions on rewards. What’s interesting is that despite not conditioning on returns during training or eval, AD still maximizes returns by imitating the learning history of the source RL algorithm which improves over time. The main contribution of the paper is showing that in-context RL emerges simply by imitating the learning histories of an in-weights RL algorithm.
> > >
> > > [1] Efficient Off-Policy Meta-Reinforcement Learning via Probabilistic Context Variables, ICML 2019 http://proceedings.mlr.press/v97/rakelly19a/rakelly19a.pdf
> > >
> > > [2] Meta-Q-Learning, ICLR 2020 https://openreview.net/pdf?id=SJeD3CEFPH
> > >
> > > [3] Recurrent Model-Free RL Can Be a Strong Baseline for Many POMDPs, ICML 2022 https://proceedings.mlr.press/v162/ni22a/ni22a.pdf
> > >
> > > [PART 3 of 3]

---

> > > > ### Comment · Reviewer_8EkG · 2022-11-18
> > > > **Regarding this review and the authors' response**
> > > >
> > > > I think this reviewer brings up some interesting and valid points, but agree with the authors that the reviewer seems to be missing the core contribution of
> > > >
> > > > > showing that in-context RL emerges if you model the learning histories of an RL algorithm with an imitation learning loss
> > > >
> > > > and that this contribution
> > > >
> > > > > is not transformer specific.
> > > >
> > > > In particular, I'm concerned by the reviewer's statements that
> > > >
> > > > > The only difference that I see with this paper and those works is use of transformer vs recurrent models to encode the context aside from online vs offline setting.
> > > >
> > > > and
> > > >
> > > > > In general, the proposed method is not original and based on previous meta-RL works. Using the decision transformer with context is the main contribution of this paper
> > > >
> > > > Since this reviewer seems to be possibly missing the core contribution of the paper, if they do not respond to the authors' response before the end of the discussion period, then I respectfully suggest that the AC (at least partially) disregard this review.
> > > >
> > > > Edit: this reviewer has engaged in the discussion (privately).  Therefore, since my suggestion to disregard this review was conditional on this reviewer being unresponsive, I do not think that the AC disregard this reviewer's review.

---

### Decision · Program_Chairs · 2023-01-20

**Decision:**

Accept: notable-top-5%

**Justification For Why Not Higher Score:**

N/A

**Justification For Why Not Lower Score:**

The fact that it works is quite surprising to me and (to another reviewer). It seems quite important to share with the community that this idea works.

**Metareview: Summary, Strengths And Weaknesses:**

The paper proposes a transformer-based imitation learning (offline RL) method that replaces the return-to-go in Decision Transformer by a long context. The key idea is that the long context contains the history of a policy improvement, and thus we can benefit more from this information than simple return-to-go scalar, e.g., for generalization. The experiment is performed very thoroughly on various tasks such as adversarial bandit, Dark Room, DMLab, etc and successfully demonstrated the claimed benefits. There are no clear reproducibility issues.

Strength. The idea is insightful and novel. The evaluation is very thorough and comprehensive. The paper is well and clearly written. The method is also simple (in a good way!) and insightful. "The fact that this approach worked was quite surprising"


Weakness. Relationship to the related works could be more clarified.


**Note From Pc:**

if the above contains the word "oral" or "spotlight" please see: "oral" presentation means -> notable-top-5% and "spotlight" means -> notable-top-25%. As stated in our emails, we are disassociating presentation type from AC recommendations